



# Spatial distribution of enhanced BrO and its relation to meteorological parameters in Arctic and Antarctic sea ice regions

Sora Seo[1], Andreas Richter[1], Anne-Marlene Blechschmidt[1], Ilias Bougoudis[1] and John Philip Burrows[1]

[1] Institute of Environmental Physics, University of Bremen, Bremen, Germany

*Correspondence to*: Sora Seo (sora.seo@iup.physik.uni-bremen.de)

**Abstract.** Satellite observations have shown large areas of elevated BrO covering several thousand $km^2$ over the Arctic and Antarctic sea ice region in polar spring. These enhancements of total BrO columns result from increases in stratospheric or tropospheric bromine amounts or both, and their occurrence may be related to local meteorological conditions. In this study, the spatial distribution of the occurrence of total BrO column enhancements and the associated changes in meteorological

parameters are investigated in both the Arctic and Antarctic regions using 10 years of GOME-2 measurements and meteorological model data. Statistical analysis of the data presents clear differences in the meteorological conditions between the 10 year mean and episodes of enhanced total BrO columns in both polar sea ice regions. These differences show pronounced spatial patterns. In general, atmospheric low pressure, cold surface air temperature, high surface-level wind speed and low tropopause heights were found during periods of enhanced total BrO columns. In addition, spatial patterns of prevailing wind

directions related to the BrO enhancements are identified in both the Arctic and Antarctic sea ice region. The relevance of the different meteorological parameters on the total BrO column is evaluated based on a Spearman rank correlation analysis, finding that tropopause height and surface air temperature have the largest correlations with the total BrO vertical column density. Our results demonstrate that specific meteorological parameters can have a major impact on the BrO enhancement in some areas, but in general, multiple meteorological parameters interact with each other in their influence on BrO columns.

## 1 Introduction

Bromine compounds play an important role in atmospheric chemistry in particular with respect to removal of ozone. It has been estimated that bromine contributes about 25 % to the global destruction of stratospheric ozone, and up to 50 % to polar stratospheric $O_3$ depletion (McElroy et al., 1986; Harder et al., 2000).

   Stratospheric bromine is predominantly present in its inorganic form, originating from both natural and anthropogenic

organic sources. Man-made brominated hydrocarbons (halons) are long-lived and are transported to the stratosphere where they release bromine atoms and Br and bromine monoxide (BrO) through UV photolysis and oxidation. Methyl bromide ($CH_3Br$) released from the ocean by natural processes and used in agriculture is oxidized in the troposphere but so slowly that it is also transported to the stratosphere (Harder et al., 2000; Theys et al., 2009b). In addition, rapid vertical transport of short-lived bromine compounds, which are mainly of natural origin, also affects the stratospheric bromine budget significantly





(Sinnhuber et al., 2005; Liang et al., 2010). In the stratosphere, organic bromine source gases are converted into inorganic forms of $Br_y$, such as Br, BrO, $BrONO_2$, HOBr, HBr, BrCl, $2Br_2$, either by direct photolysis or by reaction with OH and O (Sinnhuber et al., 2005; Theys et al., 2009b). Inorganic bromine is then involved in a variety of stratospheric chemical reactions. The ozone catalytic removal cycles involving bromine atoms, Br, and bromine monoxide (BrO), known as BrOx, have higher chain lengths than those involving chlorine atoms, and chlorine monoxide (ClO), together known as ClOx. However, there has

been much less bromine released by man than chlorine (e.g. primarily chlorofluorocarbon compounds, CFCs). Overall, the BrOx catalytic destruction cycles are significant but not as important as those of ClOx. More details of the current assessment of the role of bromine in the stratosphere are reported in the recent ozone assessment (WMO, 2018).

Tropospheric ozone depletions linked to bromine chemistry were first discovered in the Arctic ~30 years ago (Barrie et al., 1988) from the analysis of $Br^-$ on filters. The oceanic biosphere releases a variety of organobromine compounds (e.g. $CH_3Br$,

$CH_2Br_2$, $CHBr_3$ etc) as well as organochlorine compounds (e.g. $CH_3Cl$, $CH_2Cl_2$ etc) and organoiodine compounds (e.g. $CH_3I$, $CH_2I_2$ etc). It is therefore not surprising that bromine is present in the troposphere. However, as there is approximately 600 times less bromide ($Br^-$) than chloride ($Cl^-$) in the oceans it was at the time surprising to find that inorganic reactions play an important role in releasing bromine to the troposphere. An important tropospheric inorganic bromine source is the polar sea ice region (Kaleschke et al., 2004 and references therein). The rapid and sudden appearance of BrO over the polar regions has

been called the "bromine explosion" (Barrie and Platt, 1997). A large amount of active bromine compounds is released by an autocatalytic heterogeneous mechanism and leads to substantial ozone depletion (Fan and Jacob, 1992; McConnell et al., 1992; Simpson et al., 2007) and mercury depletion (Steffen et al., 2008) in the polar boundary layer in spring. Examples of the heterogeneous chain reactions involving BrO are:

$$Br_2 + hv\ (350\ nm < \lambda < 500\ nm) \rightarrow 2Br \tag{R1}$$

$$2(Br + O_3 \rightarrow BrO + O_2) \tag{R2}$$

$$2(BrO + HO_2 \rightarrow HOBr + O_2) \tag{R3}$$

$$2(HOBr_{(g)} \rightleftharpoons HOBr_{(aq)}) \tag{R4}$$

$$2(HOBr_{(aq)} + Br^-_{(aq)} + H^+_{(aq)} \rightarrow Br_{2(g)} + H_2O_{(aq)}\ ) \tag{R5}$$

$$Net:\ 2O_3 + 2HO_2 + 2Br^-_{(aq)} + 2H^+_{(aq)} \rightarrow Br_2 + 4O_2 + 2H_2O \tag{R6}$$

This is autocatalytic cycle because the $Br_2$ released is photolysed to generate Br and the chain reaction accelerates. Another autocatalytic mechanism for halogen release from sea salt aerosol starts from the reaction of HOBr with $Cl^-$ in the presence of acid as shown below (Vogt et al., 1996; Sander et al., 2003):

$$HOBr_{(aq)} + H^+_{(aq)} + Cl^-_{(aq)} \rightarrow BrCl_{(aq)} + H_2O \tag{R7}$$

$$BrCl + Br^- \rightleftharpoons Br_2Cl^- \tag{R8}$$

$$Br_2Cl^- \rightleftharpoons Br_2 + Cl^- \tag{R9}$$

$$Br_2 + hv\ (350\ nm < \lambda < 500\ nm) \rightarrow 2Br \tag{R1}$$

$$2(Br + O_3 \rightarrow BrO + O_2) \tag{R2}$$

$$2(BrO + HO_2 \rightarrow HOBr + O_2) \tag{R3}$$



$2(\text{HOBr}_{(g)} \rightleftharpoons \text{HOBr}_{(aq)})$      (R4)

Net: $2O_3 + 2HO_2 + H^+_{(aq)} + Br^-_{(aq)} \rightarrow \text{HOBr} + 4O_2 + H_2O$      (R10)

Although usually at lower concentrations than in polar regions, tropospheric bromine is also detected over salt lakes (Hebestreit et al., 1999; Hörmann et al., 2016), in volcanic plumes (Bobrowski et al., 2003; Theys et al., 2009a) and in the marine boundary layer (Sander et al., 2003; Saiz-Lopez et al., 2004).

BrO is observed by both ground-based and satellite measurements using the differential optical absorption spectroscopy
(DOAS) technique (Platt and Stutz, 2008). Ground-based measurements such as long-path DOAS (Hönninger et al., 2004; Stutz et al., 2011), multi-axis DOAS (Hönninger et al., 2004; Frieß et al., 2011), and chemical ionization mass spectrometry (Liao et al., 2011; Choi et al., 2012) provide good temporal coverage and in some cases the vertical profile of BrO, while UV-visible nadir satellite measurements allow us to study the global distribution of BrO columns with good spatial coverage. In particular, since observing BrO by satellites became possible, studies on the mechanism for large-scale release of bromine
over both polar sea ice regions, where a management of ground-based instruments is difficult, have been carried out.

The first satellite observations of polar BrO events were performed using the GOME instrument (Wagner and Platt, 1998; Richter et al., 1998; Chance, 1998). A long time-series of total BrO columns and the spatial distribution of regions with enhanced BrO in both hemispheres were investigated using 6 years of GOME data (Hollwedel et al., 2004), finding that general features of the BrO distribution are similar between years but the strength of BrO explosion events varies from year to year
with an increasing trend in both polar regions. SCIAMACHY, which was launched after GOME, can measure scattered and reflected solar radiation in limb and nadir geometry. Thus, both BrO vertical profiles and BrO column densities could be retrieved from two different observation modes of SCIAMACHY and an averaged global background of tropospheric BrO was estimated by comparing the integrated stratospheric BrO profile with the simultaneously measured total BrO column (Sinnhuber et al., 2005; Rozanov et al., 2011). OMI and GOME-2, which were launched in 2004 and 2006, respectively,
provide data with improved spatial resolution and signal to noise ratio enabling to answer additional scientific questions related to polar springtime BrO explosion events. Salawitch et al. (2010) showed that the locations of total BrO column hotspots during Arctic spring observed by OMI are coincident with high total $O_3$ column and low tropopause height, suggesting a stratospheric contribution for the BrO enhancements. Theys et al. (2011) developed and applied an algorithm to retrieve tropospheric BrO columns from GOME-2 measurements and investigated polar tropospheric BrO hotspot areas. They found
that elevated BrO columns are occasionally related to stratospheric processes due to tropopause descend in line with the result of Salawitch et al. (2010). However, they argued that the stratospheric origin cannot account for all cases and the release of bromine in the troposphere also contributes to satellite observed BrO plumes. Choi et al. (2012) conducted a comparison of satellite-derived tropospheric BrO and aircraft in-situ BrO profiles obtained from the ARCTAS and ARCPAC field campaigns for several events of rapid enhancement of BrO columns. From the two Arctic field campaigns, they showed that retrievals of
tropospheric BrO columns from OMI and GOME-2, combined with modelled stratospheric BrO estimates, show reasonably good agreement with in-situ tropospheric BrO observations. Recently, the improved spatial resolution ($3.5\text{x}7$ $km^2$) of TROPOMI onboard Sentinel-5 Precursor which was launched in October 2017 has enabled the detection of spatial variations





and small-scale emission sources in more detail. Seo et al. (2019) showed the advantage of the high spatial resolution and signal to noise ratio of TROPOMI BrO retrievals in various source regions.

Meteorological conditions for the occurrence of bromine explosion events have also been studied. Jones et al. (2006) and Jacobi et al. (2006) found that ozone depletion events and elevated BrO concentrations are related to a stable and shallow boundary layer occurring under temperature inversions and low wind speeds. Such conditions allow chemical reactions to proceed efficiently, the boundary layer acting as a closed reaction chamber. However, enhanced BrO events were also detected during episodes of high wind speed. Jones et al. (2010) found that BrO explosion events can occur under environmental

conditions consisting of high wind speeds and the presence of blowing snow. Choi et al. (2018) also showed a significant association between the temporal and spatial extent of tropospheric BrO explosions observed from OMI and GEOS-5 simulated sea salt aerosol emissions generated by blowing snow. They demonstrated that saline aerosol particles generated during blowing snow events serve as a source of reactive bromine in the bromine explosion mechanism. The role of wind speed and atmospheric stability in determining the lower tropospheric BrO vertical distribution was reported by Peterson et al. (2015)

using MAX-DOAS observations. In that study, high wind speeds were linked to some of the high columns of BrO, but based on the low frequency of these cases, they argued that high wind speeds and blowing snow are not the sole driver of halogen activation. A bromine explosion event linked to cyclone development in the Arctic was investigated by Blechschmidt et al. (2016). The vertical lifting and high wind speeds driven by the front of the polar cyclone can transport cold brine coated snow and salt aerosols into the free troposphere, resulting in bromine explosion events and extending BrO plume lifetime through

continuous supply of reactive bromine.

    As mentioned above, many studies on the possible sources of BrO enhancements and the driving meteorological conditions in polar regions have been conducted using ground-based and satellite measurements. Study results clearly indicate that meteorological conditions affect the processes of BrO enhancements in several ways. However, these previous studies focus on specific case-studies or relatively short-term data sets, which limits the investigation of favourable meteorological

conditions associated with the occurrence of enhanced BrO columns in the polar regions. In this study, to overcome this snapshot treatment of elevated BrO events during polar spring, and to obtain a more general understanding of the enhancements of total BrO columns, we statistically analyse the spatial distribution of occurrence frequency of enhanced total BrO column and its relationship to various meteorological parameters in the Arctic and Antarctic sea ice regions by using a 10 year long-term dataset. In particular, the relationship between total BrO vertical columns retrieved from GOME-2A/2B and

meteorological fields including sea level pressure, surface level wind speed and direction, surface air temperature, and tropopause height were investigated. Differences in meteorological conditions and their regional characteristics between high BrO situations and the mean field were investigated in order to better understand meteorological effects on processes involved in BrO enhancements. Finally, based on Spearman rank correlation analysis, the degree of influence of different meteorological parameters on total BrO columns was evaluated and the most important meteorological parameters, influencing BrO, and their

regional patterns were identified.



## 2 Data

### 2.1 GOME-2 BrO retrievals

The GOME-2 is a series of three identical instruments operating on board the Metop satellites. They were launched sequentially to enable continuous long-term monitoring of atmospheric composition with the same instrument specification (Callies et al.,
2000; Munro et al., 2016). The first GOME-2 aboard Metop-A was launched in October 2006, and the second and third onboard Metop-B and Metop-C were launched in September 2012 and November 2018, respectively (hereafter referenced as GOME-2A, GOME-2B and GOME-2C, respectively). GOME-2 is a nadir-viewing scanning UV-vis spectrometer with four channels covering the spectral range between 240 and 790 nm at a spectral resolution of 0.26-0.51 nm (Munro et al., 2016). The spatial resolution of GOME-2 data is typically 80x40 km$^2$ and the Metop platforms all have an equator crossing time of 09:30 local
time in the descending node (Munro et al., 2016).

In this study, we use data from GOME-2A from 2008 to 2011 and data from GOME-2B from 2013 to 2018. The first year of data after the launch of GOME-2A and GOME-2B is not used to avoid sampling bias from incomplete coverage during this time period. Also, as the swath width of GOME-2A was reduced from 1920 km to 960 km in July 2013 resulting in a change in ground pixel size to 40x40 km$^2$, we used the GOME-2B data since 2013. BrO slant column densities (SCDs) were retrieved
by applying the DOAS technique to the GOME-2 Level 1 spectra using the settings summarised in Table 1. These settings have been selected based on the results from an earlier study on BrO retrievals using data from different satellite instruments (Seo et al., 2019). The absorption cross sections used are BrO (Wilmouth et al., 1999), $O_3$ at 223 K and 243 K (Serdyuchenko et al., 2014), $NO_2$ (Vandaele et al., 1998), OClO (Kromminga et al., 2003), HCHO (Meller and Moortgat, 2000), and $O_4$ (Thalman and Volkamer, 2013). A Ring spectrum to account for rotational Raman scattering was calculated by the SCIATRAN
model (Vountas et al., 1998) and a fourth-order polynomial was included in the BrO retrieval. As background spectrum, daily averaged earthshine spectra from the equatorial Pacific were used (± 30°N, 150-240°E) and a post-processing of the BrO slant columns was performed to minimize the impact of instrumental degradation and differences between the two instruments.

When using a Pacific background spectrum, the retrieved differential slant columns need to be corrected by adding the BrO slant column over that region. Here we follow earlier studies (Richter et al., 2002; Sihler et al., 2012; Seo et al., 2019) and
assume a BrO vertical column of $V_{norm,ref} = 3.5 \times 10^{13}$ molec cm$^{-2}$ over the Pacific. The corresponding SCD is computed by multiplying with the geometric air mass factor $A_{geo}$:

$$A_{geo} = \frac{1}{\cos(SZA)} + \frac{1}{\cos(VZA)} \quad \text{(SZA: solar zenith angle, VZA: viewing zenith angle)} \tag{1}$$

As the differential BrO slant columns over the Pacific are not exactly 0, the mode μ of a Gaussian, fitted to their distribution is taken into account for the final correction:

$$SCD = DSCD - \mu + A_{geo} \cdot V_{norm,ref} \tag{2}$$

To account for differences in the light path through the atmosphere, total geometric vertical columns were derived from retrieved BrO SCDs by application of a stratospheric air mass factor (AMF), which considers only viewing geometry and wavelength. Only total BrO VCDs having solar zenith angles smaller than 85° were used in this study.





## 2.2 Sea ice data

Sea ice concentrations (Spreen et al., 2008), i.e., the percentage of a given area covered with sea ice relative to the total, retrieved from AMSR-E (Advanced Microwave Scanning Radiometer-EOS) and AMSR2 (Advanced Microwave Scanning Radiometer 2) satellite measurements were used to identify the sea ice domain for the 2008-2011 and 2013-2018 periods, respectively. The reason for the missing of analysis for 2012 in this study is that there was a gap between AMSR-E and AMSR2. AMSR-E stopped producing data in October 2011 and calibrated brightness temperature data from the AMSR2, the successor
of AMSR-E, have been released in January 2013. Sea ice concentrations are retrieved by the ARTIST Sea Ice (ASI) algorithm based on the polarization difference of brightness temperature at the 89 GHz channel and weather filtering using other channels (Spreen et al., 2008). Sea ice concentration data are provided on a daily basis with high spatial resolution of $6.25 \times 6.25$ km$^2$ by the Institute of Environmental Physics, University of Bremen (https://seaice.uni-bremen.de/sea-ice-concentration/) and these data products were used in this study.


## 2.3 Meteorological data

To explore meteorological conditions in polar regions, European Centre for Medium-Range Forecasts (ECMWF) ERA Interim reanalysis data (Dee et al., 2011) (https://apps.ecmwf.int/datasets/data/interim-full-daily/) were used. More specifically, sea level pressure, air temperature at 2 m, and surface winds at 10 m at a spatial resolution of 0.75 x 0.75 degree and a 6 hour time
resolution were extracted from the data. The tropopause height was computed from ECMWF ERA-Interim reanalysis data with 91 levels by applying the dynamical (potential vorticity) criterion of 3 PVU (1 PVU = $10^{-6}$ m$^2$ s$^{-1}$ K kg$^{-1}$) or higher to define stratospheric air (Ebojie et al., 2014).

## 3 Method

### 3.1 Data matching

To investigate the relationship between BrO and various meteorological factors over the polar sea ice regions, a spatio-temporal matching of the different datasets is required. According to previous studies, enhanced BrO columns are observed over the continent near the coast in the presence of blowing snow, and even over the interior of the continent by long-range transport of air masses from the sea ice regions (Choi et al., 2018). However, development and maintenance of large-scale enhanced
BrO plumes requires a continuous supply of reactive bromine over a large area, and the origins of these sources are typically, located on the polar sea ice. Thus, we limited the study domain for long-term analysis to the region where the sea ice remains. Since the spatial resolution of daily sea ice concentration data from AMSR-E and AMSR2 is higher than that of GOME-2 BrO columns, GOME-2 BrO data were selected if an average of sea ice concentration located within the GOME-2 satellite pixel is higher than 5 %. The selected GOME-2 BrO data over the sea ice region are also matched with ECMWF meteorological





datasets in both time and space. First, the meteorological datasets were linearly interpolated with respect to the observation time of GOME-2 BrO, and then spatially matched by interpolation with an inverse weighting proportional to the distance of the nearest four pixels from the GOME-2 BrO pixel. The temporally and spatially matched datasets were used to explore the spatial behaviour of the relationship between BrO columns and meteorological parameters in the next sections.

**3.2 Detection of enhanced total geometric BrO columns over sea ice regions**

To investigate the characteristics of occurrences of enhanced BrO, we first need to establish a reasonable detection criterion for BrO enhancements. In previous studies, satellite observations of enhanced BrO columns during spring, termed as "BrO hotspots", were defined as the region where the total column BrO is elevated by at least $2\times10^{13}$ molec cm$^{-2}$ relative to the zonal mean (Salawith et al., 2010; Theys et al., 2011). Hollwedel et al. (2004) defined the high BrO events as tropospheric BrO columns above the $5.5\times10^{13}$ molec cm$^{-2}$. In addition to these definitions, Theys et al. (2011) analysed the events of possible

stratospheric origin as the case when the retrieved tropospheric BrO column is lower than $3.5\times10^{13}$ molec cm$^{-2}$. We could detect enhanced BrO columns by applying the same threshold value and method used in previous studies. However, this study is based on 10 years of data in both hemispheres, and not on case studies or analysis of short-term data covering less than 2 years as in previous studies, and therefore a new detection criterion of enhanced BrO valid for long-term data is required.

Figure 1 shows the monthly histogram of GOME-2 total BrO VCDs over the Arctic (March and April) and Antarctic sea

ice regions (September and October) for the study period from 2008 to 2018. Average total BrO VCDs are in general normally distributed representing the mean background BrO. However, the right side of the BrO column distribution has higher frequencies of occurrence compared to the left side and is long-tailed due to BrO enhancements during polar springtime. We defined a threshold for BrO hotspot detection as the mode of the BrO column distribution $+2\sigma$ (standard deviation of the distribution) based on the monthly fitted Gaussian distribution using 10 years of GOME-2 data to have a consistent approach

for the analysis of the long-term dataset. If a total BrO column is larger than the threshold, it is identified as an occurrence of enhanced BrO. The thresholds for BrO hotspot detection are $7.64\times10^{13}$ and $7.54\times10^{13}$ molec cm$^{-2}$ for March and April, respectively, over the Arctic sea ice, and $7.50\times10^{13}$ and $7.40\times10^{13}$ molec cm$^{-2}$ for September and October, respectively, over the Antarctic sea ice (see blue dashed lines in Fig.1). Although the thresholds are in a similar range, the limits for the Arctic are slightly higher than those of the Antarctic, and the limits for the early spring of March and September are slightly higher

than those for April and October. Our statistical results are consistent with the results of Hollwedel et al. (2004), who also found that high BrO events and areas covered by BrO clouds are usually larger in the Arctic than in the Antarctic from 6 years GOME measurements.

One issue to note is that monthly thresholds for identification of BrO hotspots in each hemisphere determined by this statistical method have a limitation in discerning a clear source of enhanced total BrO column. In principle, there are three

conceivable explanations for enhanced total BrO columns observed by satellite: (1) the enhancement results from descending BrO enriched stratospheric air when the tropopause is low, (2) the increased results from bromine explosion events occurring in the troposphere, (3) the enhancements are due to a combination of both stratospheric and tropospheric contributions. Thus,



to assess whether total column BrO enhancements, detected by using the BrO VCD threshold criteria defined above, is of stratospheric or tropospheric origin, we will investigate related meteorological factors in the next sections.


## 4 Result

### 4.1 Frequencies of enhanced BrO columns and spatial distributions

To identify areas where BrO hotspots occur frequently, the spatial distribution of the frequency of enhanced BrO occurrences was investigated over a long time period. To map the spatial distribution of frequency, a reference grid is required because the

positions of the satellite pixels are not constant. In this study, a reference grid with 200x200 km resolution was used. The enhanced BrO occurrence frequency $f_{EBrO}$ was calculated by dividing the number of pixels classified as enhanced BrO column in a given reference grid cell by the total number of satellite pixels within the reference grid cell.

Figure 2 shows how frequently enhanced total BrO columns were observed over the Arctic and Antarctic sea ice regions in spring. As can be seen in Fig. 2, patterns and magnitudes of $f_{EBrO}$ are different between the Arctic and Antarctic, and also vary

between polar spring months. In the Arctic, enhanced total BrO columns are frequently observed over the north of the Canadian coast with a frequency of ~0.25 (25 %). Also, over the Hudson Bay, $f_{EBrO}$ is higher than ~0.18 in both March and April. The spatial distribution patterns of $f_{EBrO}$ are mostly similar in March and April, but relatively high values of ~0.15 are observed over the Chukchi Sea and the East Siberian Sea in April. In contrast to the Arctic, where stronger regional enhancements in BrO occurrence are evident, $f_{EBrO}$ is distributed relatively uniformly around the Antarctic continent with a comparatively low

number of local enhancements over the Weddell and Ross Sea. In addition, unlike in the Arctic, where the values of $f_{EBrO}$ are similar in March and April, the range of frequency values in Antarctica is smaller in October than in September, in particular, around the Weddell and Ross Sea. Statistical analysis using decadal GOME-2 observations showed spatial and temporal behaviours of total BrO column enhancements in the Arctic and Antarctic region. Spatial variations in the occurrence frequency of enhanced total BrO columns may be linked to the influence of various meteorological parameters, and this will be discussed

in detail in Section 4.2.

### 4.2 Relationship between enhanced BrO columns and meteorological parameters

In this section, we examine the relationship between the occurrence of enhanced total BrO columns and various meteorological conditions, focusing on the magnitude and spatial distribution of changes in meteorological conditions when comparing situations with BrO enhancements to the mean.

### 4.2.1 Sea level pressure

The first parameter investigated is sea level pressure. Figure 3 shows histograms of ERA-interim sea level pressure data from 2008 to 2018 for the Arctic and Antarctic sea ice region during spring (March and April for the Arctic, September and October



for the Antarctic). The red line shows the frequency distribution of sea level pressure for cases with enhanced total BrO, whereas the black line represents the frequency distribution of the mean field using all sea level pressure data for the study

period. As can be seen in Fig. 3, the frequency distribution of sea level pressure is shifted towards lower sea level pressure during BrO enhancements in both polar regions, indicating that enhancement in total BrO vertical column is related to lower sea level pressure. This decrease of sea level pressure can also be clearly seen from a comparison of the mean sea level pressure map for all measurements with the mean sea level pressure map for the enhanced BrO cases (see Fig. 4). When total BrO columns are enhanced, sea level pressure is generally decreased by up to ~25 hPa in most areas of the Arctic when compared

to the average sea level pressure. Significant decreases in sea level pressure are in particular found in the central Arctic and the Bering Strait. Sea level pressure tends to increase slightly only over the Kara Sea in March. Similar to the results in the Arctic, lower sea level pressure is associated with enhanced BrO columns in Antarctica (see Fig. 5). Negative sea level pressure anomalies are found in most areas, and a particularly pronounced negative anomaly in sea level pressure ($\sim 20$ hPa) is observed over the east side of the Antarctic Peninsula in September. Also, from Fig. 5, it is clear that the decrease of Antarctic sea level

pressure for the enhanced BrO cases in October is not as large as in September.

Our analysis using long-term datasets demonstrates that enhancements in total BrO columns are associated with negative sea level pressure anomalies in both the Arctic and Antarctic sea ice region. The Arctic and Antarctic have fundamentally different geographical features. The Arctic is a frozen ocean surrounded by land whereas Antarctica is a frozen continent surrounded by ocean, which leads to differences in the major synoptic pressure systems. The lower tropospheric circulation

over the frozen Arctic ocean is dominated by high pressure systems over the continents, while that of the sea ice zone around Antarctica is driven by a strong low pressure belt (Jones et al., 2010; Screen et al., 2018). Although the atmospheric dynamic systems are different between the Arctic and Antarctic sea ice regions, sea level pressure is generally lower during the enhancement of BrO columns in both polar regions, which indicates that atmospheric depressions have an influence on generating enhanced total BrO.

Previous studies support the association between atmospheric low pressure systems and enhanced BrO columns. Jones et al. (2010) found that large scale tropospheric ozone depletion events and enhanced BrO columns appear around large low pressure systems in the Antarctic region using data from tethersondes, free-flying ozonesondes, and satellites. Blechschmidt et al. (2016) revealed a link between polar cyclones and bromine explosion events development for a case over the Arctic. These studies argue that vertical lifting and high wind speeds in synoptic scale atmospheric low pressure systems result in development of

blowing snow, so that the BrO explosion reaction cycle occurs around the wind-blown brine coated snow particles in tropospheric air. Convergence and ascent of air occur along fronts within low pressure systems. This convective process facilitates air masses potentially having reactive bromine source conditions at the ground to be transported to higher altitudes and cooled. Theys et al. (2009) and Salawitch et al. (2010) showed that some of the enhancement in total BrO columns are associated with increases in stratospheric BrO due to a decrease of the tropopause height, coincident with low pressure systems.

The details of how surface level wind and tropopause height, which can be affected by changes in atmospheric pressure, are associated with the enhancement of the total BrO column will be discussed in later sections.





### 4.2.2 Surface level temperature

The relationship between air temperature at 2 m and the enhancements of total BrO columns was investigated in the same way as for pressure in the previous section. From Fig. 6, it can be seen that the surface level air temperature is low during

enhancements of total BrO column in both the Arctic and Antarctic sea ice regions. The air temperature frequency distribution of the Antarctic is shifted more clearly towards lower temperatures than that of the Arctic. The highest frequencies of 27 % in the mean field and 32 % in the enhanced BrO field are detected at the same temperature range of -20 to -15 °C in the Arctic. However, for the Antarctic, the temperature range of the highest frequency is -20 to -15 °C in the high BrO situations, compared to -15 to -10 °C in the mean field. Spatial distributions of monthly temperature differences for the Arctic and Antarctic are

also shown in Fig. 7 and 8, respectively. One remarkable feature in these maps is that temperature differences vary depending on region. When the total BrO column is enhanced, negative surface level air temperature anomalies (up to ~12°C colder compared to the long-term mean) are found over the Arctic sea ice region except for the central Arctic region where positive anomalies occur. The pattern of surface level air temperature in the Antarctic is similar to that of the Arctic as shown in Fig. 8. Atmospheric temperature during enhanced BrO events is lower in most areas of the Antarctic and a significant decrease of

temperature is found at the margin of the sea ice. In contrast, a slight increase in temperature is associated with the enhancement of BrO around the Antarctic coastal region. These patterns indicate that not only relative temperature changes but also absolute temperatures are linked to the conditions in which enhanced BrO events can occur. Negative temperature anomalies are needed for BrO enhancement in the marginal sea ice regions where temperatures are closer to the freezing point than in the central Arctic and Antarctic, where temperatures are always low.

Temperature effects in the chemical mechanism of bromine release were discussed in several previous studies. Sander et al. (2006) demonstrated why bromine release is accelerated on cold saline surfaces using a 1-dimensional atmospheric chemistry model. They found that the acid-catalyzed atmospheric bromine explosion cycle is triggered in their simulations by precipitation of carbonates at a temperature below 263 K leading to reduced buffering capacity of the alkaline sea water and facilitating its acidification. Model calculations identified the strong temperature dependency of the equilibrium reactions (R8)

and (R9) releasing $Br_2$ and BrCl. These reactions transform the bromide in brine into reactive bromine. The cold conditions associated with bromine explosions at the surface are similar to those required to form frost flowers in polar regions. Frost flowers which are water ice, coated with brine can act as a primary source of bromine explosion events. They are mainly formed when open polynyas or leads freeze at very low temperature (Martin et al., 1996; Kaleschke et al., 2004; Obbard et al., 2009). Kaleschke et al (2004) reported that potential frost flower regions, associated with cold surface air temperature, match

spatially the source regions of enhanced BrO columns detected by satellites. The results from our long-term statistical analysis are consistent with those from previous studies showing the importance of low temperature conditions for bromine explosions. In summary, the present results combined with findings on chemical mechanisms, assumed to be responsible for the development of tropospheric bromine explosion events in previous studies, indicate that atmospheric temperature is one of the important parameters in the BrO column variability.



### 4.2.3 Surface level wind speed and direction

Next, surface level wind speed is investigated to evaluate how this may affect the occurrence of total BrO column enhancements. Figure 9 shows the frequency distribution of wind speed at 10 m for the average field and for enhanced BrO cases of the 10 years of measurements in the Arctic and Antarctic sea ice region. The distribution is shifted towards high wind speeds in both polar regions for enhanced total BrO vertical columns, the increase in wind speed being more pronounced in the Antarctic region. The difference in wind speeds is also confirmed by the spatial distribution maps (Fig. 10 and 11). Higher wind speeds are observed in most Arctic and Antarctic regions for situations with enhanced total BrO columns. In particular, differences in wind speed of more than 5 m·s$^{-1}$ and high wind speeds of over 10 m·s$^{-1}$ are found at specific regions of the Arctic such as the eastern coast of Greenland, the Bering Strait and the central Arctic. In the Antarctic region, wind speeds greater than 12 m·s$^{-1}$ are predominantly observed over the sea ice margins and some of the Antarctic coastline. Our results show that enhancements of BrO columns are mainly related to positive wind speed anomalies.

The relationship between surface wind speed and enhanced BrO column has been discussed in previous studies, and still is under debate. Some of the related studies showed tropospheric ozone depletion events and bromine explosion events at lower wind speeds of less than 8 m·s$^{-1}$ in polar sea ice regions (Simpson et al., 2007; Jones et al., 2010). These case study results support the opinion that tropospheric BrO explosion events occur efficiently at low wind speeds since under these conditions, reactants can accumulate within the stable boundary layer while under strong wind conditions, a dilution of bromine sources due to rapid vertical mixing and horizontal spreading prevents the BrO explosion. On the other hand, highly saline sea salt aerosols produced during blowing snow events driven by high surface wind speeds can act, if sufficiently cold, as a bromine source in the tropospheric BrO explosion mechanism (Yang et al., 2008; Obbard et al., 2009; Jones et al., 2009; Blechschmidt et al., 2016). Yang et al. (2010) successfully simulated bromine explosion events using a global chemistry transport model by considering bromine emissions from sea salt aerosols and saline snow lying on sea ice during blowing snow events. Also, Jones et al (2009) and Blechschmidt et al. (2016) reported that high wind speeds and vertical lifting caused by a cyclone can provide blowing snow. When the wind speed exceeds ~8 m·s$^{-1}$, snow particles begin to dislodge from the surface, and at wind speeds above ~12 m·s$^{-1}$, active mixing and transport of snow and sea salt aerosols within the boundary layer become possible (Jones et al., 2009). Thus, blowing snow created by high surface wind speeds is a plausible mechanism releasing reactive bromine sources into the boundary layer or above it. This supports the notion that high wind speeds also lead to enhanced BrO in the lower atmosphere.

The spatial distribution map of surface wind speed anomalies derived in this study shows that during the enhancement of total BrO vertical columns, wind speeds are generally enhanced. However, the average wind speed field during the high BrO cases shows values of 6-8 m·s$^{-1}$ in most areas. For the tropospheric bromine explosion events created by strong winds, previous studies indicate that wind speeds above ~12 m·s$^{-1}$ are required. The regions that satisfy this wind speed threshold consistently are confined to the Bering Strait, the central Arctic, and the east coast of Greenland in the Arctic and the Antarctic sea ice margins and some coastal locations. This behaviour is clearly identified in the spatial distribution maps of the relative frequency





of high wind speeds for the occurrence of enhanced total BrO column (see Fig. 12) which show where strong surface winds contribute to the enhancement of BrO columns. In Fig. 12, high frequencies above 30 % are found over the central Arctic and eastern coast of Greenland in the Arctic, whereas in the Antarctic, they are most frequently detected around the marginal ice zone of the Weddell and Ross Sea. In short, the results shown in Fig. 10-12 suggest that the occurrence of enhanced BrO columns in the corresponding regions where wind speeds are high and positive wind speed anomalies appear may be significantly associated with a tropospheric bromine source generated by high wind speeds.

In order to investigate whether not only the wind speed but also the wind direction affects the occurrence of BrO enhancement in terms of regions, the relative frequency of wind direction for data with enhanced BrO columns was mapped. The relative frequency was calculated for the case that the number of data collected within the reference grid is greater than 20 to avoid errors from using too small sample size. The wind direction was divided into eight groups at intervals of 45 degrees. From Fig. 13, we can identify which wind directions are related to BrO enhancements and the regional characteristics of these prevailing wind directions. First, the northern and western wind directions are more frequent compared to the southerly and easterly winds when enhancements of total BrO column occur in the Arctic sea ice region. High occurrence frequencies of above 50 % are found around the Bering Strait and the eastern coast of Greenland. This is expected because the northerly winds blowing from the interior of the Arctic sea ice are likely to be cold and contain bromine sources such as saline snow, frost flowers and sea salt aerosols rather than the southerly winds from the open sea. In contrast to the prevailing northerly winds around the Arctic sea ice margins, southern winds including S, SE and SW are mainly observed around the central Arctic and high latitude regions for the BrO event cases. This may be explained by the transport of air masses with enhanced BrO from sea ice margins to high latitudes by southerly winds, which is also in line with the temperature increase pattern over the central Arctic reported in the previous section. Overall, north-westerly winds are the most dominant wind direction, associated with the enhancement of total BrO column in the Arctic, especially high occurrence frequencies being found over the northern coast of Canada, Hudson Bay and Baffin Bay. These regions have been mentioned in many previous studies as areas where satellite BrO hotspots frequently appear during Arctic spring (Salawitch et al., 2010; Theys et al., 2011; Nghiem et al., 2012). Fig. 14 shows the degree of changes in the frequency of wind directions in cases of enhanced BrO compared to the mean field in the Arctic in terms of spatial variation. In general, the south and east winds have decreased frequencies in most regions during the occurrence of enhanced BrO compared to the mean, whereas the frequencies of NW, W, and N, the main wind directions associated with BrO enhancements, increase by more than 20 %, especially around the sea ice margin area.

The spatial distribution of the wind direction frequency during the occurrence of enhanced total BrO columns is clearer in the Antarctic, compared to the Arctic. As shown in Fig. 15, the northern winds show low relative frequency overall in the Antarctic sea ice region for BrO enhancement cases, which indicates that the northerly winds have a low association with BrO enhancements. This is consistent with our understanding of the conditions required for BrO enhancements because the northerly winds blowing from the open water to Antarctica are usually relatively warm, and thus sea salt aerosols are not cold enough to trigger the bromine explosion mechanism. Another feature of wind direction distribution in the Antarctic is that





easterly winds are strongly related to enhanced BrO columns along the coast of Antarctica, whereas westerly winds are prevailing over the sea ice region except for the Antarctic coastlines. Basically, the main winds of the Antarctic can be divided into two types: (1) large-scale circulations composed of westerly winds and (2) local katabatic winds deflected in the cross-

slope direction with an eastward component due to the Coriolis force. Consequently, the predominant wind direction during the BrO enhancement also follows the Antarctic large-scale circulation. Depending on the large-scale atmospheric system, easterly winds in the coastal regions and westerly winds in the sea ice regions are dominant, but the influence of the south wind increases in the presence of enhanced BrO columns. Maps of differences in the frequency of the wind direction between the high BrO situation and the mean field (see Fig. 16) clearly show that occurrence frequencies of wind directions from

southerly and westerly directions increase during the occurrence of enhanced BrO columns. In particular, the southwestern winds prevail for situations with enhanced BrO columns, as frequencies increase by more than 20 % over a large area of Antarctic sea ice including the marginal ice zones.

We investigated further the impact of high wind speed on the total BrO column enhancement at each wind direction in the Arctic and Antarctic regions during spring. Table 2 summarizes the relative frequency of wind direction for the mean field,

for the enhanced BrO cases and for enhanced BrO cases accompanied by high wind speeds ($v \geq 12$ m·s$^{-1}$ for the Arctic and $v \geq 14$ m·s$^{-1}$ for the Antarctic), respectively. The average frequency of wind direction is generally evenly distributed between 10.6-14.7 % in the Arctic. However, for the cases with enhanced total BrO columns, the relative frequency of the northwest, west and north winds increase to 22.0, 17.9 and 16.0 %, and when limited to high surface wind speed cases, frequencies increase further to 24.7, 21.1 and 18.9 %, respectively. In the Antarctic sea ice region, the difference in the frequency between

the individual wind directions is larger than in the Arctic. The average frequency of the northeast wind is 8.1%, while the frequency of the west wind is as high as 20.1 %. The difference in the relative frequency distribution of the wind directions becomes even larger if cases with enhanced BrO are considered. The southwest and west winds are then prevailing with a relative frequency of 30.4 and 26.0 %, which increases further to 35.5 and 32.6 % under high wind speed conditions. Through the statistical analysis, we confirmed that the specific wind directions affect the BrO enhancement and if the relevant winds

become strong, the supply of reactive bromine source in the troposphere becomes active, which can contribute further to the increase of total BrO column density.

### 4.2.4 Tropopause height

The last factor investigated in connection with the enhancement of total BrO columns is the tropopause height. The relationship between the tropopause height and BrO hotspots observed from satellites was discussed in previous studies using data for

periods from several days up to two years. Salawitch et al. (2010) found that enhanced total BrO columns over the Hudson Bay observed by OMI are coincident with a low tropopause of ~5 km and high total O$_3$ column of ~450 DU. In general, the stratospheric BrO columns are anti-correlated with the tropopause height field, and thus they concluded that the elevated total BrO columns at low tropopause heights over the Hudson Bay may be attributed to an increase of stratospheric BrO columns. Theys et al. (2011) also investigated total, stratospheric and tropospheric BrO columns retrieved from GOME-2 measurements



and the corresponding tropopause heights for the northern high latitudes in spring. They found two different cases through their analysis using two years of data: (1) situations where the spatial distribution patterns of enhanced total BrO columns and low tropopause heights are consistent, (2) BrO hotspots, which are not associated with low tropopause heights and high stratospheric BrO columns. The first case shows that the increase in total BrO columns can be affected by the stratospheric contribution in line with the findings of Salawitch et al. (2010), but the second indicates that elevated total BrO columns can

also be linked to the increase of tropospheric BrO columns.

In this study, the relationship between tropopause height and total BrO column enhancements is investigated in both the Arctic and Antarctic sea ice region using 10 years long-term data in terms of magnitude, region and time, and the results are compared with those from previous studies. The relative frequency distributions of tropopause height for the mean field and the cases with enhanced total BrO columns in the Arctic and Antarctic are presented in Fig. 17. In both polar regions, the

frequency distribution is shifted towards lower tropopause height during the occurrence of enhanced total BrO columns, and this effect is larger in the Antarctic than the Arctic. This indicates that the decrease of tropopause heights is associated with the enhancement of total BrO columns, which is consistent with previous study results. Our results also show that the enhancement of total BrO columns due to tropopause descends is more prominent in the Antarctic than in the Arctic. This conclusion is different from the finding of Theys et al. (2010), who reported that the effect of low tropopause height on the

increase of BrO column seems to be more important in the Arctic than in the Antarctic region. One possible explanation for the difference apart from the different time periods investigated is that here, the frequency distribution of the tropopause height was investigated for situations with enhanced total BrO column whereas Theys et al. (2010) investigated the impact of tropopause height on the tropospheric BrO column.

In addition to the frequency distribution of tropopause height, also the differences in maps of tropopause height between

situations with enhanced total BrO column and the mean were investigated (Fig. 18 and 19). The mean field of tropopause height shows generally a tropopause height of 9.0-9.5 km range over the Arctic except for the north coast of Canada (e.g. Canadian Archipelago), Fram Strait and the Barents Sea where slightly lower tropopause heights of 8.0-8.5 km appear. For the BrO enhancement cases, tropopauses are lower in the 7.5-9.0 km range in most areas. In particular, large differences in tropopause height of ~2 km are observed over Bering Strait, Hudson Bay, and Baffin Bay, indicating that enhancements of

total BrO columns in these areas may be more affected by stratospheric BrO through tropopause descent. Similar to the Arctic case, the tropopause height is lower by up to ~2.3 km in most of the Antarctic sea ice region in situations with enhanced total BrO columns. Especially, large differences are detected around the Antarctic Peninsula in September. Negative tropopause height anomalies influence the total BrO column through the increase of the stratospheric column, or by mixing stratospheric air with high BrO concentrations into the troposphere. However, they are also associated with changes in meteorological

conditions in the troposphere, which favour BrO release close to the surface. Consequently, not only the surface level meteorological factors such as surface level atmospheric temperature, wind speed and direction but also the impact of the stratospheric field should be considered in the interpretation of enhancements in the total BrO column.





### 4.3 Correlation analysis between total BrO vertical column and meteorological parameters

To assess statistically the dependence between total BrO vertical column and multiple meteorological parameters investigated in the previous sections, we performed a Spearman's rank correlation analysis. The Spearman correlation determines the strength and direction of the monotonic relationship between two variables. Thus, this method is less sensitive to strong outliers and distribution type than the Pearson correlation, which is the reason for using the Spearman rank correlation analysis in this study.

Monthly Spearman correlation coefficients between total BrO vertical column density and meteorological parameters including sea level pressure, surface level air temperature, surface level wind speed, and tropopause height are provided in Table 3. Correlation coefficients with p-values less than 0.001, indicating that results are statistically significant, are presented here. Correlations are low for all four factors investigated (apart from tropopause heights for September in the Antarctic which shows a moderate correlation of 0.4). Overall, tropopause height shows the largest negative correlation with total BrO vertical

column density in both the Arctic and Antarctic. Total BrO vertical column is more negatively correlated with tropopause height in the Antarctic than the Arctic, and the negative correlation coefficients of March and September are larger than those of April and October. Total BrO vertical column has also a negative correlation with the surface level air temperature in both polar regions, but these correlation coefficients are smaller than correlation coefficients with tropopause height. Larger correlation coefficients are found in March and September than April and October. Sea level pressure has a negative correlation

of -0.23 with total BrO column in September in the Antarctic region, but correlations are even lower in the Arctic and in other months. Surface wind speed is not significantly correlated with total BrO vertical column density in either the Arctic or Antarctic. This could either be due to stratospheric air dominating the total BrO column or due to the large variability of wind speed conditions under which tropospheric BrO explosion events occur according to previous studies described above.

      The regional difference of the statistical dependence between total BrO vertical column and meteorological factors were

also investigated by performing the Spearman correlation analysis for each cell of the reference grid. Figure 20 displays the spatial distribution of Spearman correlation coefficients between total BrO vertical column and each meteorological factor where the p-value < 0.001. The strong negative correlation between total BrO vertical column and tropopause height appears in both the Arctic and Antarctic region. The strongest negative correlation coefficients reaching values below -0.6 are found over the Hudson Bay, Baffin Bay and Bering Strait in the Arctic sea ice region, while the regional variation in the correlation

is not large in the Antarctic region. These strong negative correlations between total BrO vertical columns and tropopause heights during springtime indicate that the total BrO vertical column tends to increase when the tropopause height is lowered, which means that the contribution of stratospheric origins to total BrO columns is significant. It was also confirmed from the spatial distribution of correlation coefficient that the influence of tropopause height on the total BrO vertical column varies depending on the region. Over the Hudson Bay and Bering Strait with stronger negative correlations, the stratospheric BrO

fields associated with the change of tropopause height may have a greater impact on total BrO columns than in other regions.



Surface level air temperature also shows different Spearman correlation patterns depending on the region. The correlation between total BrO vertical column and temperature is negative at relatively lower latitudes of the Arctic and then turns to positive values over the central Arctic region. This indicates that an increase of surface air temperature is related to the enhancement of total BrO vertical column over the central Arctic, while a decrease of temperature is associated with the increase of total BrO column in most areas except for the central Arctic. In contrast to the Arctic, opposite correlation patterns between total BrO column and surface air temperature as a function of latitude are not detected in the Antarctic sea ice region. Correlations are not very significant around the Antarctic coastal region and negative correlation coefficients of ~ -0.3 are found in most sea ice areas. These spatial distribution patterns of correlation coefficients for surface air temperature are similar to those of the difference in the temperature between the enhanced total BrO field and the mean field (see Fig. 7 and 8).

Spatially, sea level pressure is negatively correlated with total BrO vertical column in most regions of the Arctic and Antarctic. However, sea level pressure has weak negative correlations with total BrO column compared to the values found for tropopause height and surface air temperature. Wind speed is positively correlated with total BrO vertical column in most areas of both polar sea ice, but the correlation is very weak with lrl < 0.2.

The correlation analysis using the long-term dataset improves our understanding of the influence of meteorological factors on total BrO vertical columns in terms of the region and the time of year (month). The strongest influence of the different parameters on total BrO column, having a large correlation coefficient, is tropopause height, which indicates that the stratospheric contribution is significant in the total BrO column density variations. This is presumably because low tropopause heights result in a larger contribution of stratospheric BrO column to the total column. Consequently, accurate stratospheric correction is important in studying the tropospheric bromine explosion events and estimating tropospheric BrO content from the measured total BrO columns. Among the surface level meteorological parameters, air temperature impacts on the total BrO column density, arguably because temperature is an important factor in the chemical mechanism of reactive bromine release in the lower atmospheric layer. Sea level pressure and surface level wind speed are negatively and positively correlated with the total BrO vertical column density, but correlations are lower than for surface air temperature, which indicates that their influence on total BrO column is not as large as surface air temperature.

## 5 Summary and conclusions

Bromine monoxide is located in both the stratosphere and the troposphere and large-scale BrO column enhancement is frequently observed in the Arctic and Antarctic sea ice region during polar springtime by satellites. In this study, we analysed 10 years of GOME-2 total BrO columns and various meteorological parameters to establish statistical connections between where the enhanced BrO columns mainly appear and the underlying meteorological conditions. The occurrence of enhanced total BrO columns showed regional characteristics. Relatively high occurrence frequencies are detected over the north Canadian coast, the Hudson Bay and the east Siberian Sea in the Arctic, while in the Antarctic, enhanced BrO columns are





often observed across the Weddell and Ross Sea, especially in September. The occurrence frequency of enhanced total BrO columns showed more spatial variation in the Arctic, whereas it varied more temporally in the Antarctic region.

Several meteorological parameters such as sea level pressure, surface level air temperature, wind speed and direction, and tropopause height were investigated to assess any significant relationships with the occurrence of enhanced total BrO columns. If the mean meteorological conditions are compared with those during enhanced BrO events, the latter are associated with low pressure systems, cold air temperature, high surface wind speed and a decrease of tropopause height in both the Arctic and Antarctic sea ice region. Low pressure systems can drive vertical uplifting and high wind speeds, lifting considerable amounts

of saline snow or aerosols acting as a source of reactive bromine in the troposphere. In the case of temperature, surface air temperature is clearly lower during high BrO events in most of both polar regions, but it is slightly higher over the central Arctic and the Antarctic coastal region. The slight positive temperature anomalies in the central Arctic region during BrO events may be influenced by transport of BrO rich air with the southern wind blowing from the sea ice margins where temperatures are relatively higher. Surface wind speed is generally higher during BrO column enhancements, and in particular,

high surface wind speed above 12 m·s$^{-1}$ which can drive blowing snow events is found over the eastern coast of Greenland, Bering Strait, the central Arctic and the Antarctic sea ice margins. Regional characteristics, comprising the prevailing wind direction and the wind speed during the BrO enhancement, were also identified. The occurrence of enhanced total BrO columns is closely associated with winds from the northwest, west and north in the Arctic sea ice region, whereas the dominant wind direction in the Antarctic during BrO enhancements is from the southwest and south. As characteristics of the spatial

distribution of the dominant winds associated with BrO enhancements, north-western winds are frequently observed at the north coast of Canada and over the Hudson Bay, and northerly winds over the east coast of Greenland and the Bering Strait during Arctic spring. For the Antarctic sea ice region, prevailing southwestern winds are mainly found during high BrO periods, especially in the marginal ice zones with relative frequencies larger than 50 %. Tropopause heights are significantly lower during enhancements of total BrO columns in both the Arctic and Antarctic region in agreement with earlier studies. The effect

of increased stratospheric contributions due to sinking of BrO enriched air as the tropopause descends leads to the enhancement of total BrO columns.

    We also performed a Spearman rank correlation analysis between total BrO vertical column density and meteorological factors to assess the relevance of these factors for total BrO column enhancements. Total BrO vertical column density has the strongest negative correlation with tropopause height in both the Arctic and Antarctic regions, reflecting both the importance

of contributions in the stratospheric BrO column for the total BrO column and of the link between low tropopause height and tropospheric conditions required for bromine explosions. The next most statistically significant factor is surface air temperature. One remarkable point is that the temperature is negatively correlated with total BrO column in most sea ice regions, but has a positive correlation over the central Arctic. The opposite correlation pattern in the central Arctic where surface air temperature is low might be due to the transport of enhanced BrO plumes from relatively low latitude sea ice regions and low surface

temperatures enough to form frost flowers that act as a source of bromine explosion. Sea level pressure and surface level wind





speed are negatively and positively correlated with the total BrO vertical column density, respectively, but their correlation coefficients are low and not statistically significant.

This study has focused on a statistical analysis of a large set of GOME-2 BrO total columns to derive spatial and temporal patterns of links between meteorological conditions and the occurrence of enhanced BrO columns. The results show systematic

connections between all of the parameters studied and BrO enhancements. However, such links do not necessarily constitute a cause and effect relationship, in particular as quantities such as surface pressure, wind speed, temperature and tropopause height are closely linked to each other. Another important aspect not covered by the approach of this study is the transport of air masses with enhanced BrO levels away from the region of initial bromine activation – in such cases, the correlation between meteorological parameters such as high wind speed and elevated BrO is not linked to the initial bromine release mechanism.

In future studies, other parameters such as sea ice type, snow cover or the presence of polynyas should also be included and ideally an optimal stratospheric correction applied to the BrO columns to better focus on tropospheric bromine explosions.

**Author contributions**

SS carried out the GOME-2 BrO retrievals, collected the model meteorological data, performed the analysis, and wrote the

paper. AR, JPB, AMB and IB provided helpful idea and feedback in designing of the study. AR developed the DOAS retrieval code and supported the satellite retrieval. JPB, AMB, and IB supported data interpretation. All co-authors contributed to the writing of the paper.

**Acknowledgements**

We gratefully acknowledge the funding by the Deutsche Forschungsgemeinschaft (DFG, German Research Foundation) - Projektnummer 268020496 - TRR 172, within the Transregional Collaborative Research Center "ArctiC Amplification: Climate Relevant Atmospheric and SurfaCe Processes, and Feedback Mechanisms (AC)³" in subproject C03. GOME-2 Lv1 data were provided by EUMETSAT. ERA Interim meteorological data were provided by ECMWF. We thank the sea ice remote sensing group of Institute of Environmental Physics, University of Bremen, for providing sea ice concentration data
retrieved from AMSR-E and AMSR2 (https://seaice.uni-bremen.de/start/).



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



**Table 1.** Summary of DOAS settings used for the GOME-2 BrO slant column retrievals.

| Parameter | Description |
| --- | --- |
| Fitting window | 334.6-358 nm |
| Absorption cross-sections | BrO (Wilmouth et al., 1999), 228 K<br>$O_3$ (Serdyuchenko et al., 2014), 223 and 243K<br>$NO_2$ (Vandaele et al., 1998), 220 K<br>OClO (Kromminga et al., 2003), 213 K<br>HCHO (Meller and Moortgat, 2000), 298 K<br>$O_4$ (Thalman and Volkamer, 2013), 293 K |
| Ring effect | Ring cross section calculated by SCIATRAN model |
| Polynomial | 5 coeff |
| Solar reference spectrum | Kurucz solar spectrum (Chance and Kurucz, 2010) |
| Background spectrum | Daily averaged earthshine spectrum in equatorial Pacific region |
| Intensity offset correction | Linear offset |





**Table 2.** Relative frequency (expressed in %) of surface level wind directions for all data, the enhanced BrO case and the enhanced BrO with high wind speeds for the Arctic and Antarctic in spring.

| | Wind direction (%) | | | | | | | |
|---|---|---|---|---|---|---|---|---|
| | **N** | **NE** | **E** | **SE** | **S** | **SW** | **W** | **NW** |
| **Arctic** | | | | | | | | |
| Relative frequency for all data | 14.7 | 12.6 | 12.7 | 12.5 | 11.4 | 10.6 | 11.5 | 14.0 |
| Relative frequency during the enhanced BrO occurrence | 16.0 | 11.0 | 9.2 | 7.4 | 6.7 | 9.8 | 17.9 | 22.0 |
| Relative frequency during the enhanced BrO occurrence with high wind speeds ($v \geq 12$ m·s$^{-1}$) | 18.9 | 7.3 | 7.1 | 4.5 | 4.4 | 12.0 | 21.1 | 24.7 |
| **Antarctic** | | | | | | | | |
| Relative frequency for all data | 8.4 | 8.1 | 11.4 | 10.1 | 10.4 | 18.4 | 20.1 | 13.1 |
| Relative frequency during the enhanced BrO occurrence | 2.7 | 3.0 | 6.7 | 10.4 | 13.5 | 30.4 | 26.0 | 7.3 |
| Relative frequency during the enhanced BrO occurrence with high wind speeds ($v \geq 14$ m·s$^{-1}$) | 0.5 | 1.3 | 6.6 | 9.4 | 9.4 | 35.5 | 32.6 | 4.7 |



**Table 3.** Spearman rank correlation coefficients between total BrO VCDs and four meteorological parameters. The results are shown separately for different months in spring in the Arctic and Antarctic. Note that all Spearman's rank correlation coefficients are significant ($p < 0.001$). Abbreviation: MA (March to April), SO (September to October)

|  | Arctic | | | Antarctic | | |
|---|---|---|---|---|---|---|
|  | **Mar** | **Apr** | **MA** | **Sep** | **Oct** | **SO** |
| **Sea level pressure** | -0.032 | -0.098 | -0.070 | -0.228 | -0.053 | -0.130 |
| **Temperature** | -0.242 | -0.149 | -0.180 | -0.305 | -0.110 | -0.193 |
| **Wind speed** | -0.011 | 0.067 | 0.035 | 0.038 | 0.043 | 0.040 |
| **Tropopause height** | -0.336 | -0.298 | -0.315 | -0.400 | -0.307 | -0.345 |





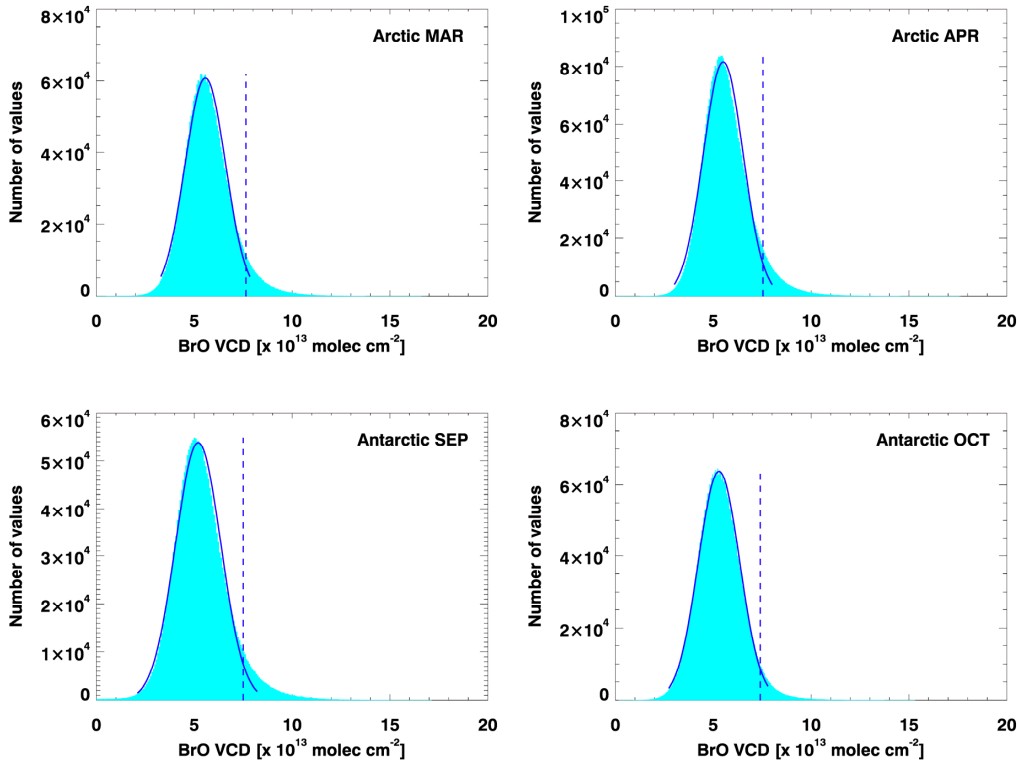


**Figure 1.** Monthly histograms of the total BrO VCD from GOME-2 measurements during the study period of 2008-2018 (March and April for the Arctic, September and October for the Antarctic). The blue solid line is a fitted Gaussian distribution and the blue dashed line indicates a range of mode+2σ (used as the monthly threshold for classifying cases of enhanced BrO columns) for the Gaussian distribution.


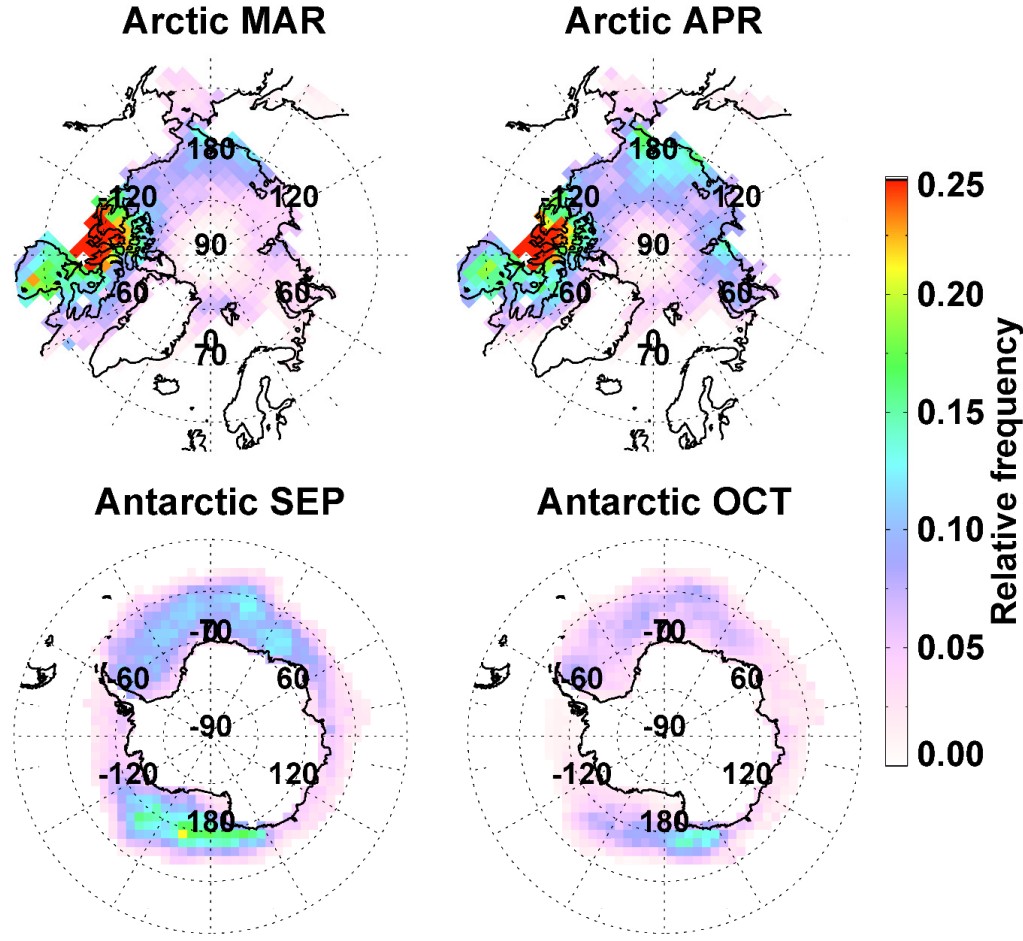


**Figure 2.** Monthly spatial distribution of the occurrence frequency of enhanced total BrO columns over the Arctic (top left: March, top right: April) and Antarctic sea ice region (bottom left: September, bottom right: October) during the study period of 2008-2018.





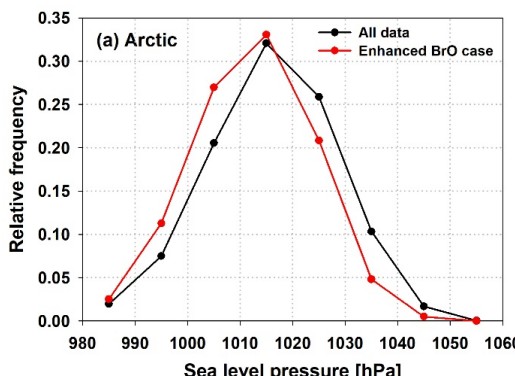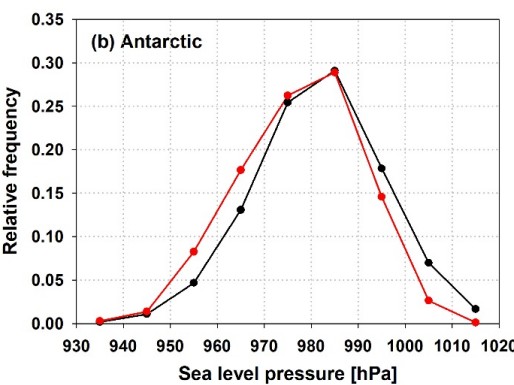

**Figure 3.** Frequency distribution of sea level pressure for all data (black line) and situations with enhanced BrO columns (red line) in (a) the Arctic and (b) Antarctic sea ice region in spring (Arctic: March to April, Antarctic: September and October).





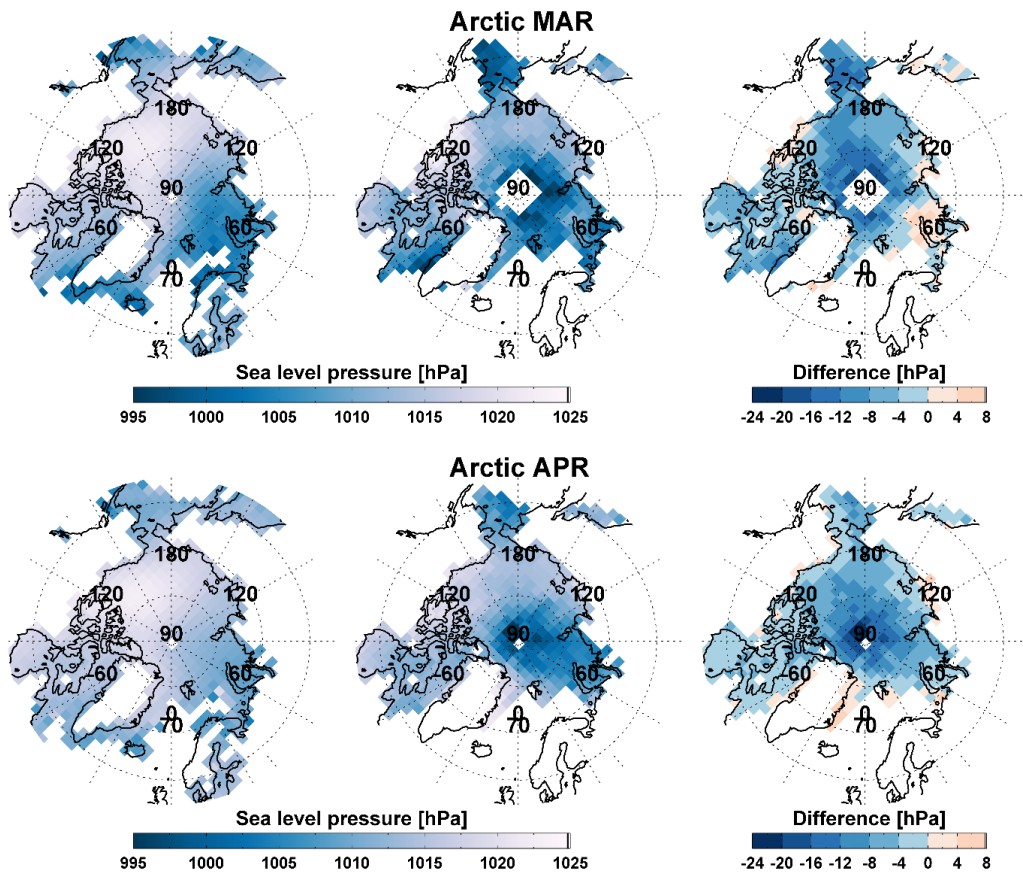


**Figure 4.** Monthly sea level pressure for the mean field (left), the enhanced BrO case (middle), and sea level pressure anomalies (difference of sea level pressure between the enhanced BrO case and the mean field) (right) over the Arctic in March (upper panel) and April (lower panel).



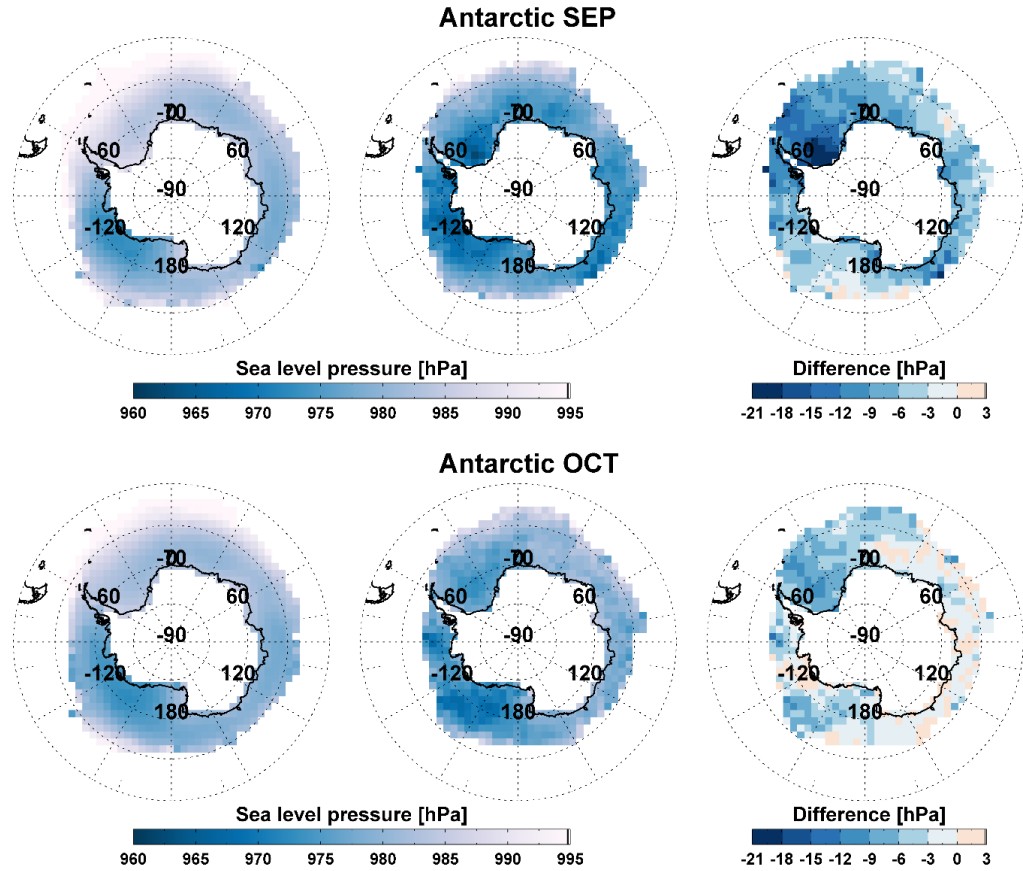


**Figure 5.** As Fig. 4, but for the Antarctic in September (upper panel) and October (lower panel).





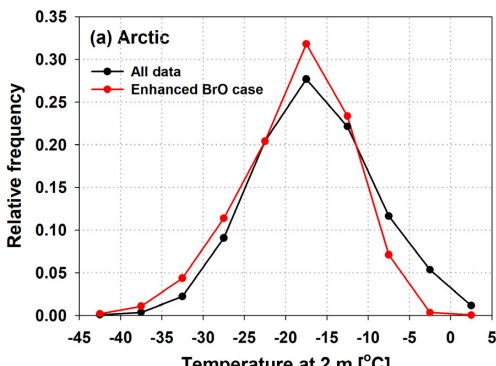
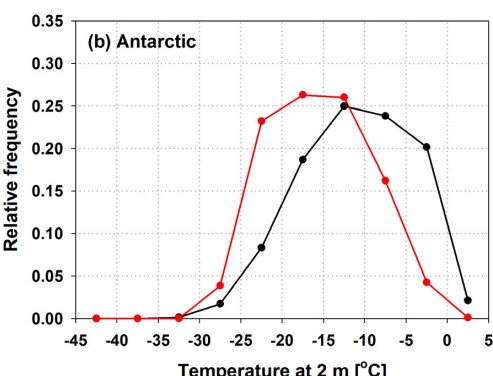

**Figure 6.** Frequency distribution of surface air temperature for all data (black line) and the enhanced BrO case (red line) in (a) the Arctic and (b) Antarctic sea ice region in spring (Arctic: March to April, Antarctic: September and October).



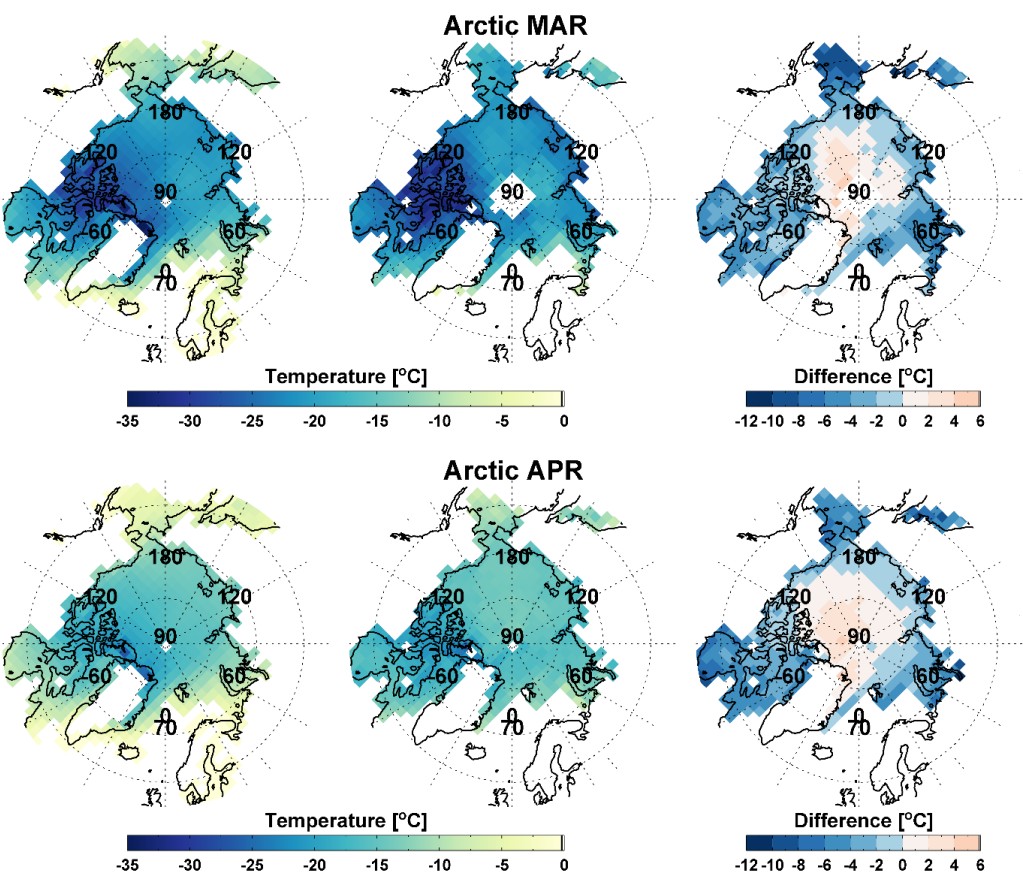

**Figure 7.** Monthly surface air temperature for the mean field (left), the enhanced BrO case (middle), and surface air temperature anomalies (difference of surface air temperature between the enhanced BrO case and the mean field) (right) over the Arctic in March (upper panel) and April (lower panel).





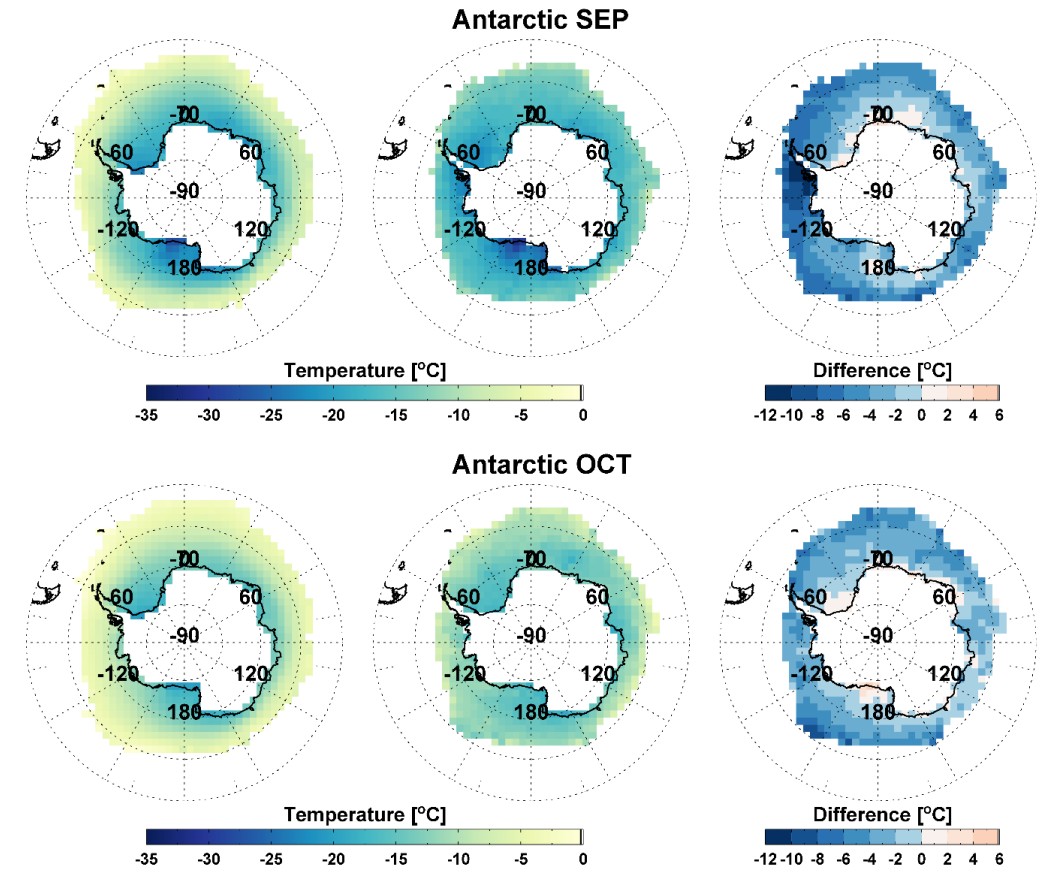


**Figure 8.** As Fig. 7, but for the Antarctic in September (upper panel) and October (lower panel).





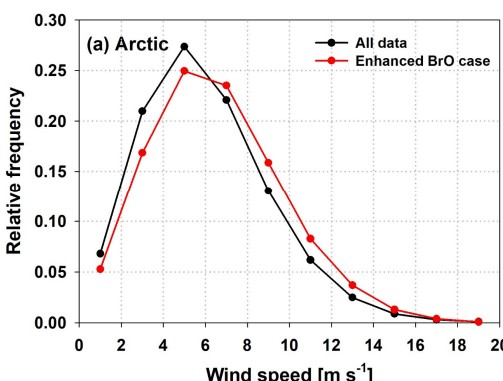 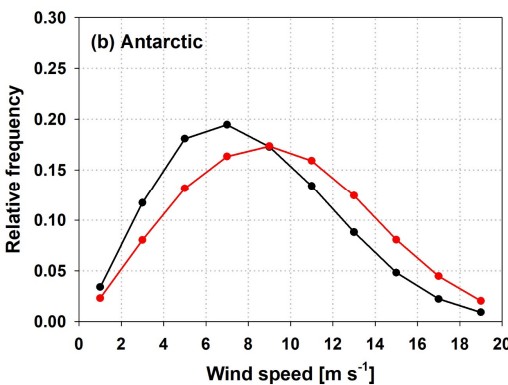


**Figure 9.** Frequency distribution of wind speed at 10 m for all data (black line) and the enhanced BrO case (red line) in (a) the Arctic and (b) Antarctic sea ice region in spring (Arctic: March to April, Antarctic: September to October).





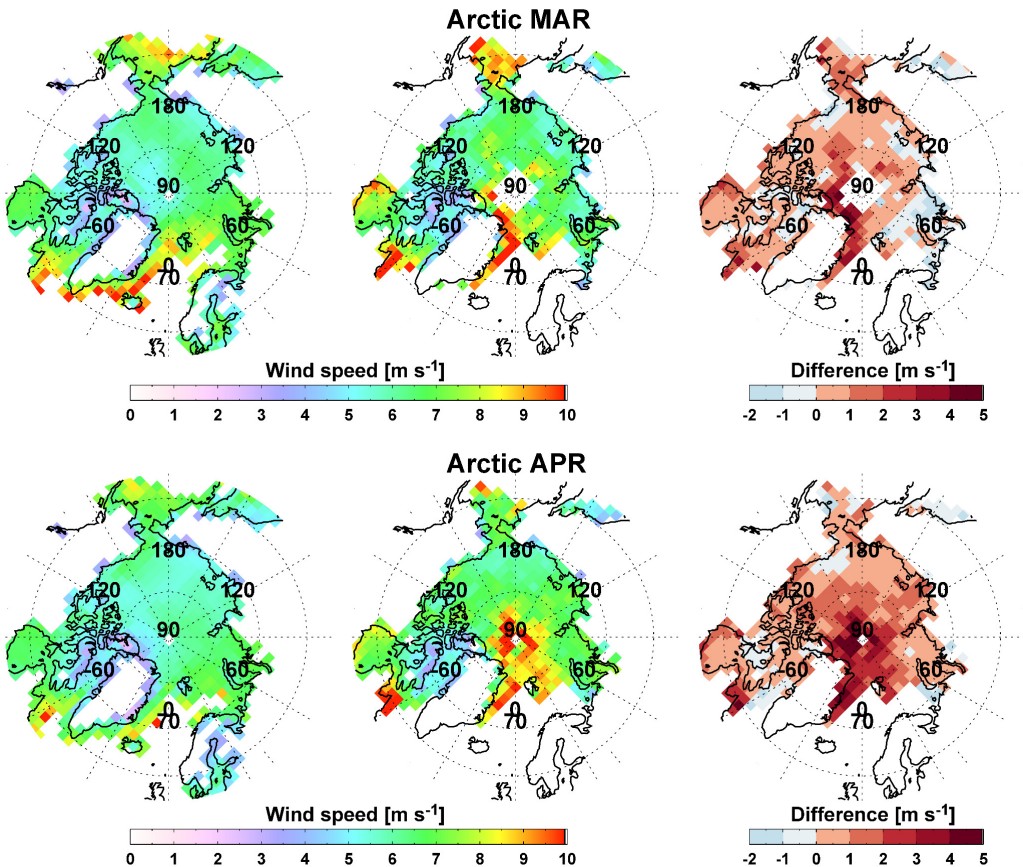


**Figure 10.** Monthly wind speed at 10 m for the mean field (left), the enhanced BrO case (middle), and surface wind speed anomalies (difference of wind speed between the enhanced BrO case and the mean field) (right) over the Arctic in March (upper panel) and April (lower panel).





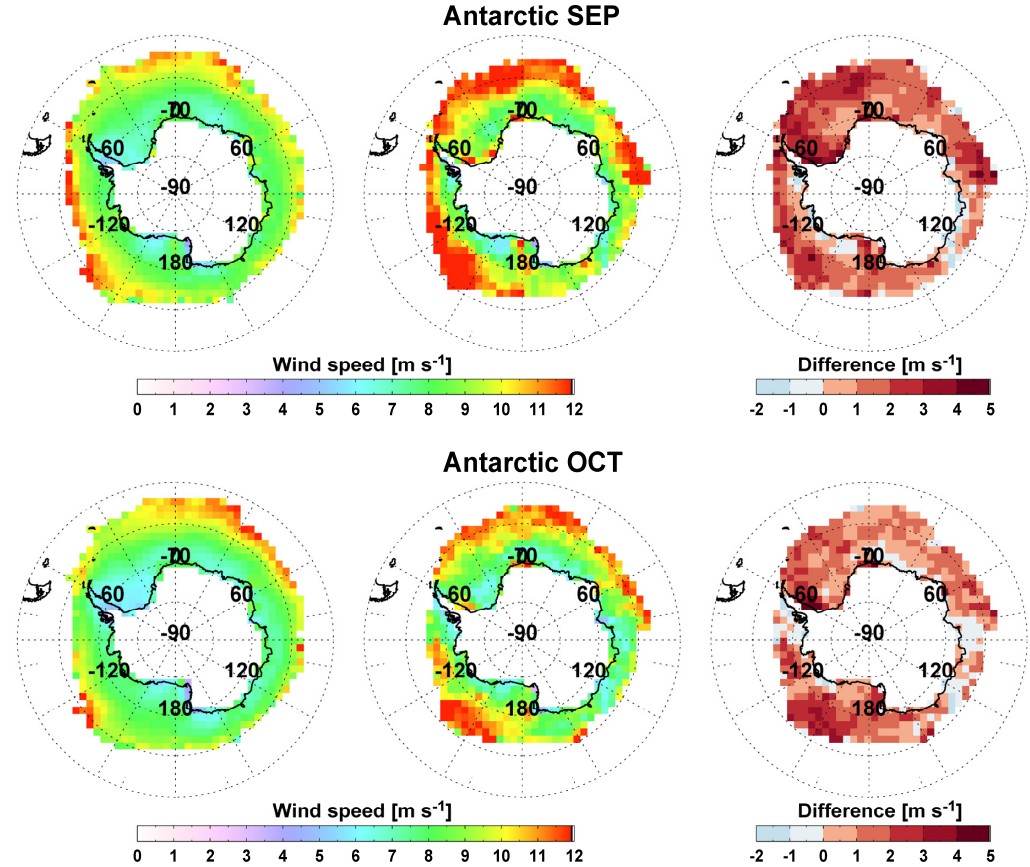


**Figure 11.** As Fig. 10, but for the Antarctic in September (upper panel) and October (lower panel).





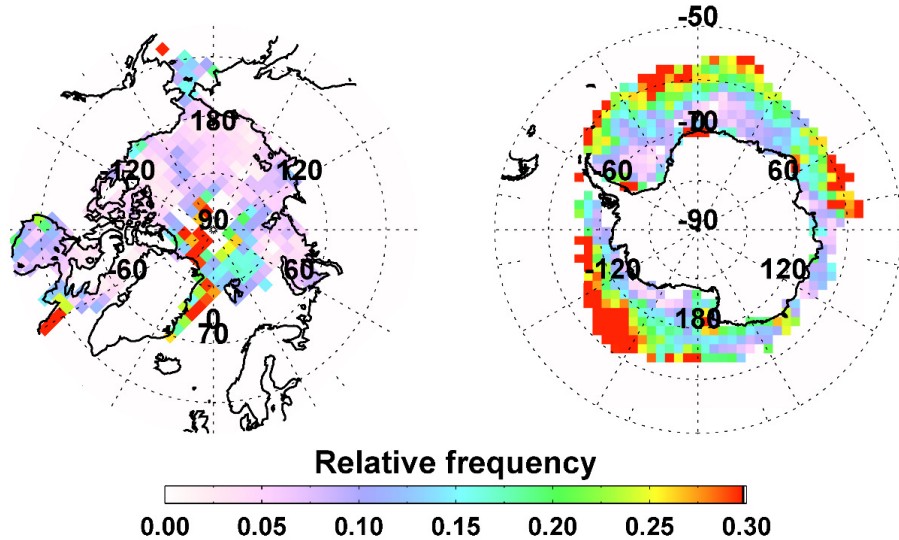


**Figure 12.** Spatial distributions of the relative frequency of high wind speeds during the enhanced BrO occurrences in the Arctic ($v \geq 12$ m·s$^{-1}$) and Antarctic ($v \geq 14$ m·s$^{-1}$) sea ice region in spring (Arctic: March to April, Antarctic: September to October).





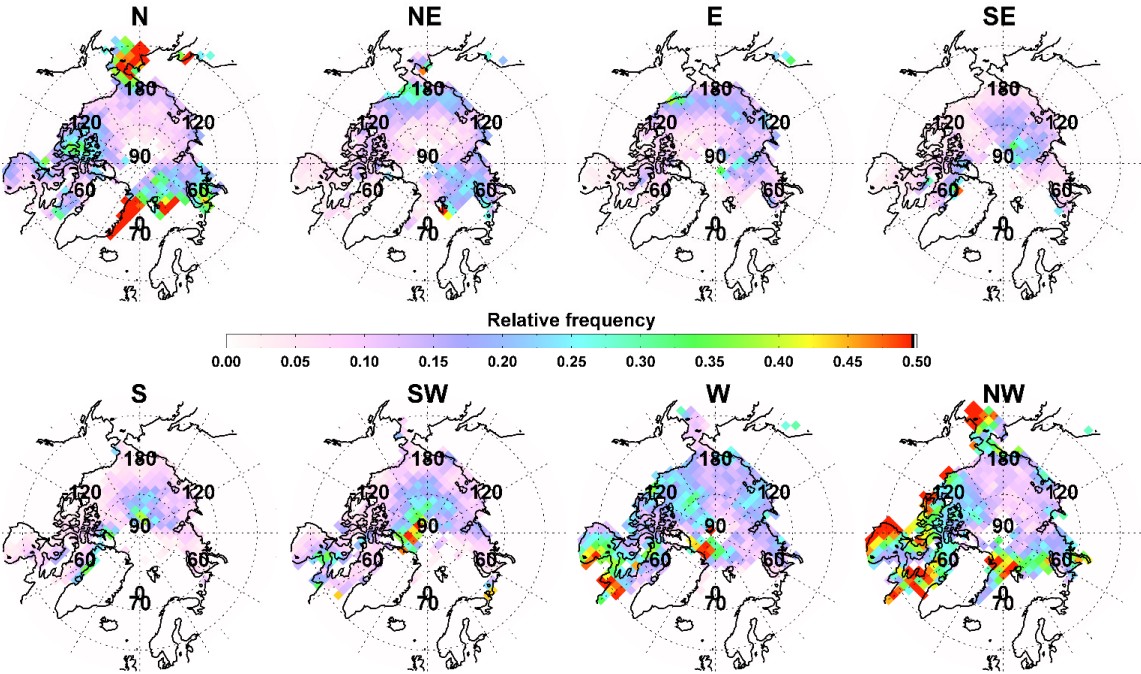

**Figure 13.** Relative frequency maps of surface level wind direction for data with BrO enhancements over the Arctic during spring (March to April in 2008-2018). The frequency was calculated for the cases where the number of data collected within the reference grid is greater than 20. The results are shown separately for different wind directions.



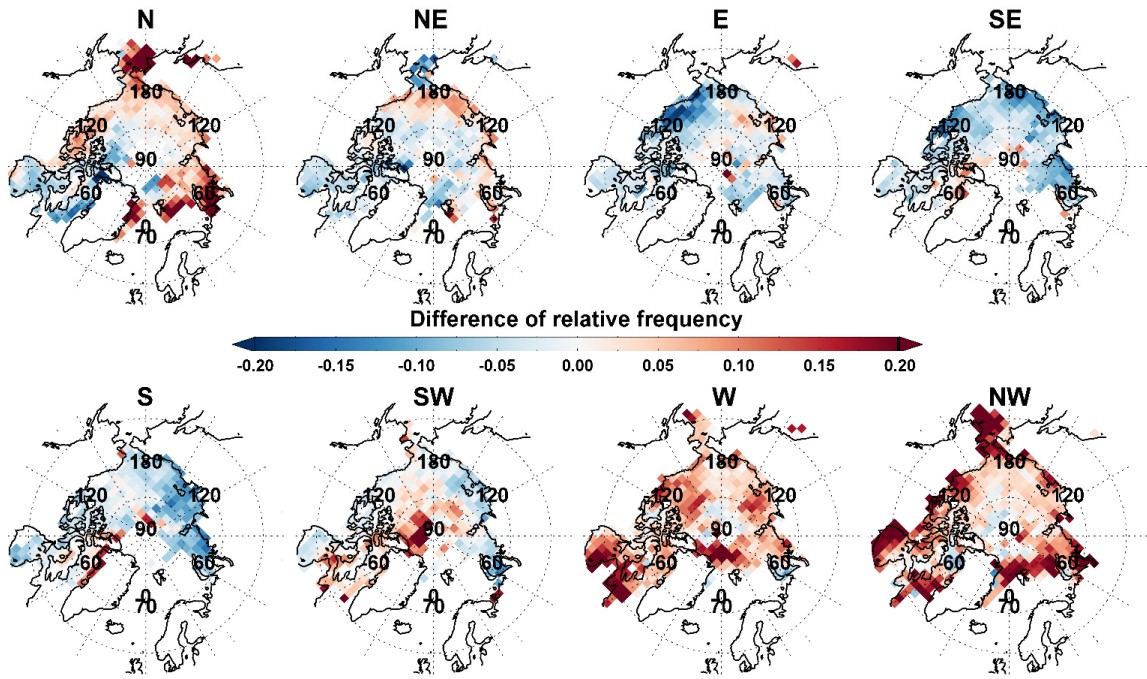

**Figure 14.** Spatial distributions of the differences in relative frequencies of wind direction between enhanced BrO cases and the mean field for the Arctic in spring (March to April in 2008-2018). The results are shown separately for different wind directions.





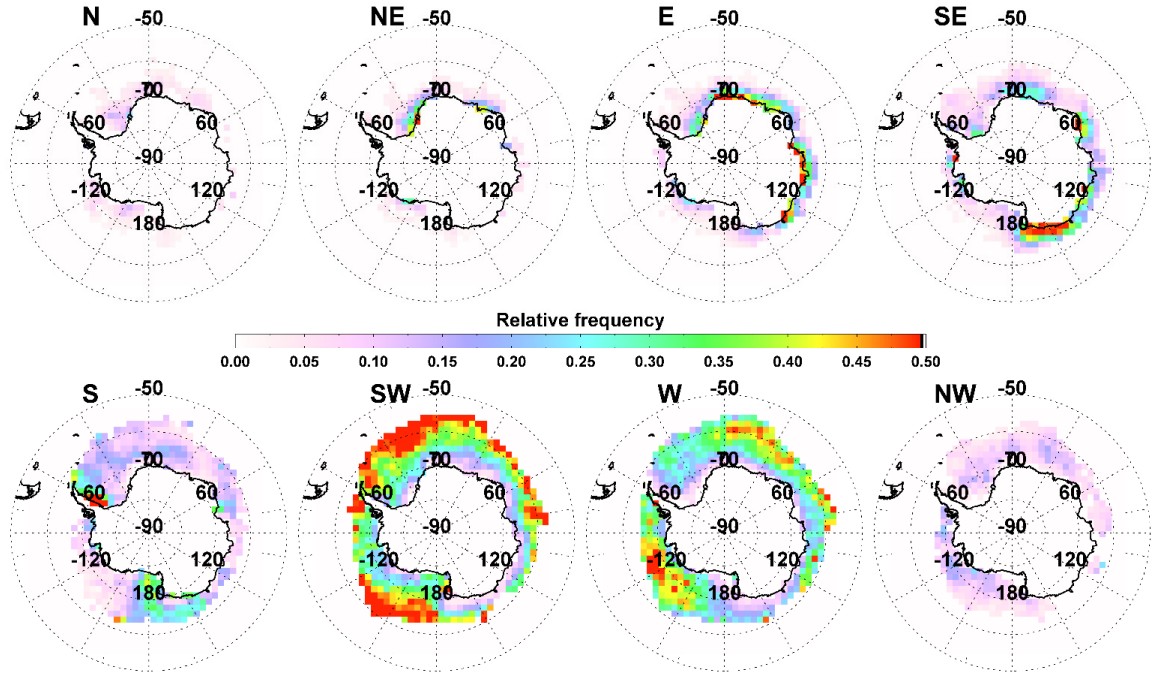


**Figure 15.** As Fig. 13, but for the Antarctic during spring (September to October in 2008-2018).





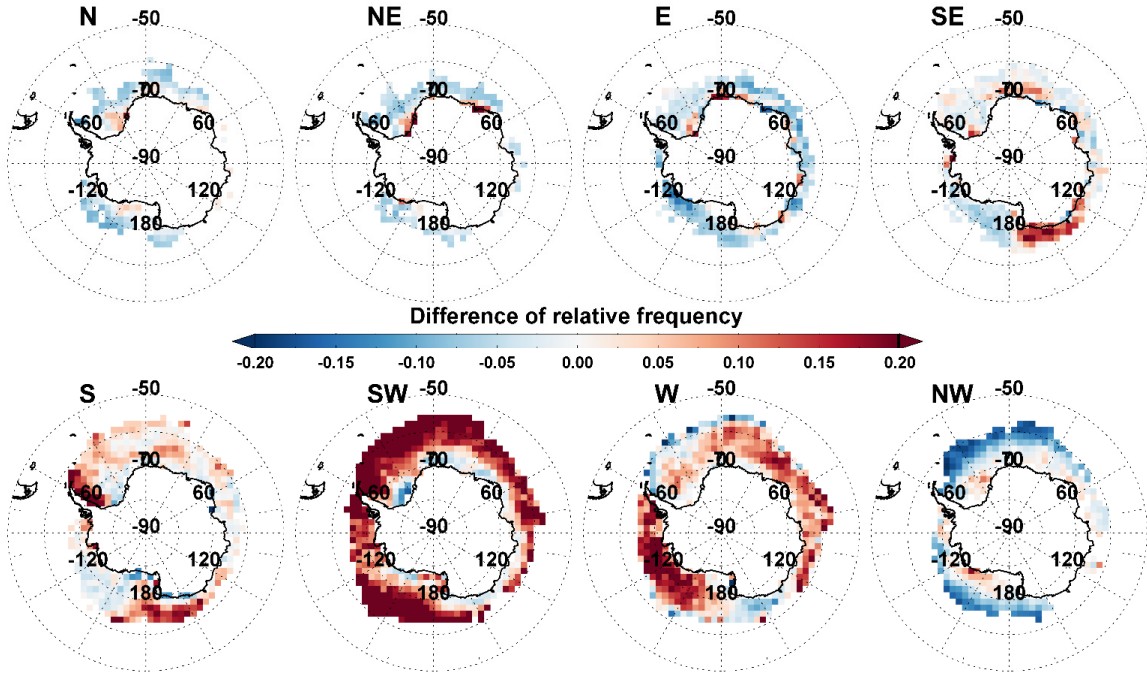

**Figure 16.** As Fig. 14, but for the Antarctic during spring (September to October in 2008-2018).





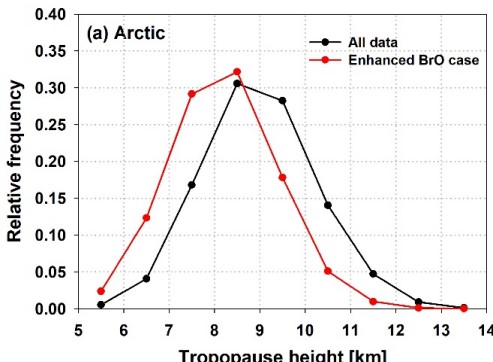 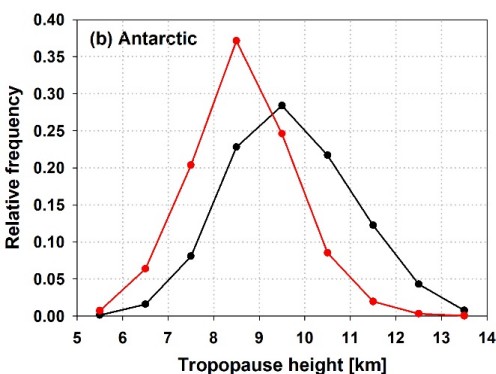


**Figure 17.** Frequency distribution of tropopause height for all data (black line) and the enhanced BrO case (red line) in (a) the Arctic and (b) Antarctic sea ice region in spring (Arctic: March to April, Antarctic: September to October).





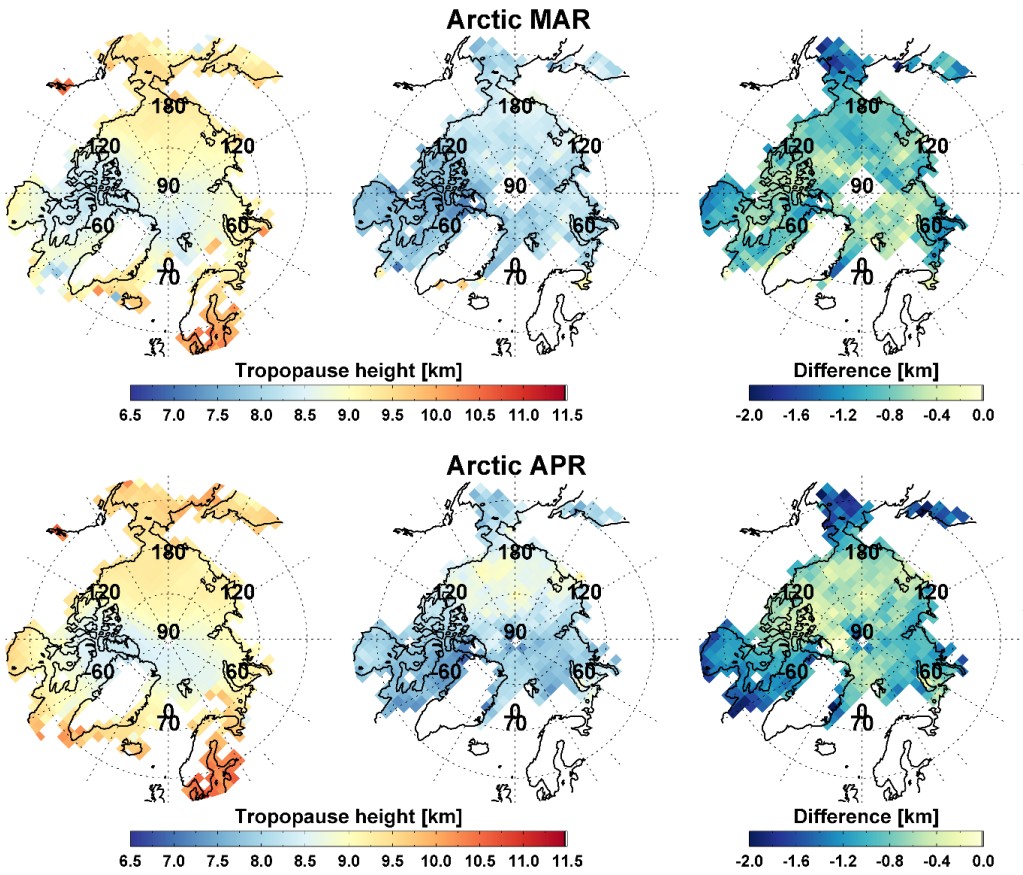


**Figure 18.** Monthly tropopause height for the mean field (left), the enhanced BrO case (middle), and tropopause anomalies (difference of tropopause height between the enhanced BrO case and the mean field) (right) over the Arctic in March (upper panel) and April (lower panel).





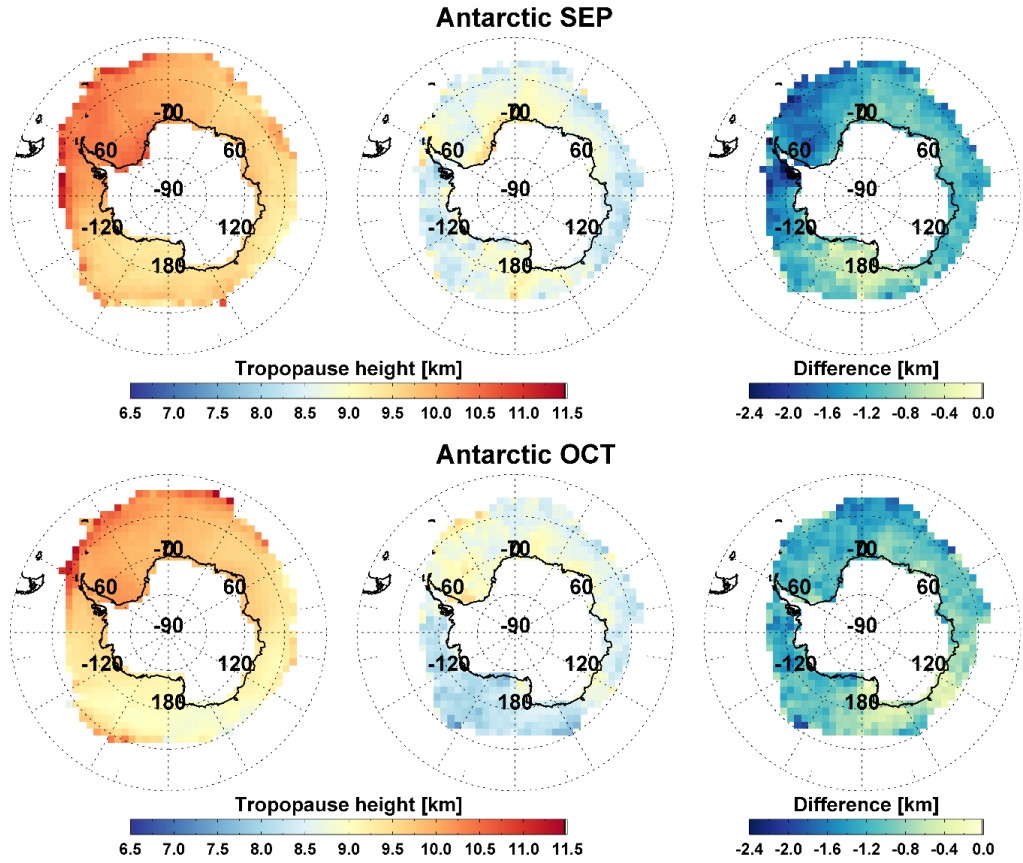

**Figure 19.** As Fig. 18, but for the Antarctic in September (upper panel) and October (lower panel).




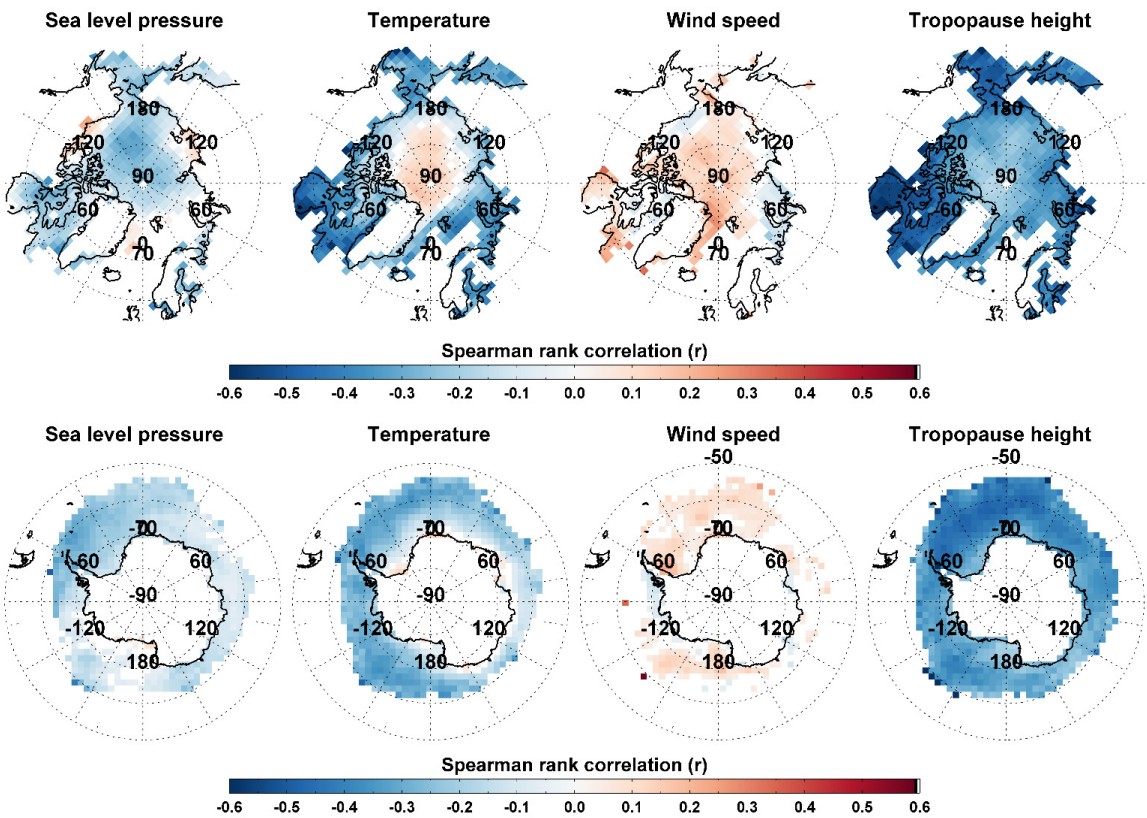

**Figure 20.** Spatial distributions of the Spearman correlation coefficients between total BrO VCD and four meteorological parameters (sea level pressure, surface air temperature, wind speed at 10 m, and tropopause height) for the Arctic and Antarctic in spring. Spearman correlation coefficients with p-value < 0.001 are only plotted.