# Peer review of "Spatial distribution of enhanced BrO and its relation to meteorological parameters in Arctic and Antarctic sea ice regions"

_Atmospheric Chemistry and Physics, 2019_

## Referee Comment (RC1) · Anonymous Referee #2 · 10 Feb 2020

**Comment to Seo et al.'s manuscript titled 'Spatial distribution of enhanced BrO and its relation to meteorological parameters in Arctic and Antarctic sea ice regions'**

General comments:

This manuscript reports a statistic analysis of polar total column BrO and its corresponding meteorological parameters (including temperature, surface pressure, wind seed, direction and tropopause height), using 10-year GOME-2 BrO and meteorological data. A further Spearman rank correlation analysis was applied to assess the dependence of total column BrO on various meteorological parameters. Some interesting results regarding the spatial distribution of enhanced BrO and its relation to the above meteorological parameters are reported which is welcome. The authors also attempted to link the relationship to relevant physical/chemical mechanism(s) being proposed previously. This study indeed adds some new knowledge to help a better understanding of the underlying processes involved. However, I found their attempt to evaluate some relevant mechanisms is dangerous and inappropriate. This is because the bromine source is a complex function of multiple (rather than one) factors and the relations between them are largely non-linear, rather than linear. For example, the SSA production from blowing snow is a complex function of wind speed, temperature and relative humidity (Yang et al., 2019), more specifically, the SSA production (and bromine flux) is actually proportional to the sublimation flux of blowing snow, rather than individual parameters (winds, temperature and *RH*). Secondly, these meteorological parameters examined are not independent, instead they are deeply correlated (also pointed by the authors). For example, they are deeply correlated during a storm system. For this reason, I suggest the authors add a table to show the cross-correlation among these meteorological parameters during enhanced BrO cases and discuss their implications to deriving your conclusions. Thirdly, it is not correct to assume that a higher correlation coefficient between two factors than others necessary means this link is stronger than other links. Unfortunately, this assumption seems being used in this manuscript to make some conclusions. For the above reasons, I suggest a major revision before recommend it to publish in ACP.

Specific comments:

P2L35-36: 'Overall, the BrOx catalytic destruction cycles are significant but not as important as those of ClOx', though this statement is correct, recent modelling-based work indicates the cross-halogen reactions (e.g. between Br and Cl) are important in terms of stratospheric ozone depletion, see recent work by Fraiser et al. (2019).

P2L43-44: 'An important tropospheric inorganic bromine source is the polar sea ice region (Kaleschke et al., 2004 and references therein)'. It would be better to cite a review paper (e.g. Abbatt et al., 2012) here, as Kaleschke et al.'s paper specifically addressed the importance of frost flowers. Or you can specify various proposed candidates of sea ice sourced bromine from frost flowers (Kaleschke et al. 2004), first-year sea-ice (Simpson et al., 2007b; Pöhler, et al., 2010), sea salt aerosol produced from blowing snow (Yang et al., 2008), stratospheric origin (Salawitch et al., 2010), snowpack photochemistry (e.g. Pratt et al., 2013) and sea spray from open leads (e.g. Peterson et al., 2015). See references shown below.

P4L119-123: 'In this study, to overcome this snapshot treatment of elevated BrO events during polar spring, and to obtain a more general understanding of the enhancements of total BrO columns, we …' I do not agree that snapshot study is less-advantaged comparing to general statistic study. They both have their individual advantage and disadvantage, they may

compensate each other at some point. I suggest a slight change to the tone used in the statement.

P15/P18: in P15L476-477 'Surface wind speed is not significantly correlated with total BrO vertical column density in either the Arctic or Antarctic.' A similar statement is also in the summary (P18L557). However, they are contrary to the results shown in Table 3 and Fig. 20, where it shows 'Note that all Spearman's rank correlations are significant (p<0.001)'. Is surface wind speed significantly correlated with total BrO vertical column density or not?

P17L533-4 and L552-555, the two statements are actually duplicated, either delete one or move one to section 4.3.2.

P27 Table 3: given that the occurrence of blowing snow has a threshold wind speed, e.g. 7~8 m/s, I suggest add an extra row in the Table to show the correlation coefficients for wind speeds > 8 m/s (or >12 m/s).

Minors:

P1L26: change 'they release bromine atoms and Br and bromine monoxide (BrO)…' to 'they release bromine atoms, e.g. Br and bromine monoxide (BrO) …'

P1L28: Theys et al., 2009b should be 3009a, as Theys et al., 2009a has not been cited.

P2L31: change '$2Br_2$' to '$Br_2$'

P2L32: again you cited 'They et al., 2009b' before their 2009a paper.

P3L67: Theys et al., 2009a should be cited before 2009b.

P4L112: 'A bromine explosion event linked to cyclone development in the Arctic was investigated by Blechshmidt et al. (2016).' The same event was also studied by Zhao et al., 2016.

Reference:

Yang, X., Frey, M. M., Rhodes, R. H., Norris, S. J., Brooks, I. M., Anderson, P. S., Nishimura, K., Jones, A. E., and Wolff, E. W.: Sea salt aerosol production via sublimating wind-blown saline snow particles over sea ice: parameterizations and relevant microphysical mechanisms, Atmos. Chem. Phys., 19, 8407–8424, https://doi.org/10.5194/acp-19-8407-2019, 2019.

Fraser, D., J., Keeble, O., Morgenstern, G., Zeng, G., A. N., Luke, X. Yang, Improvements to stratospheric chemistry scheme in the UM-UKCA (v10.7) model: solar cycle and heterogeneous reactions. *Geoscientific Model Development*, 12. 1227-1239. 10.5194/gmd-12-1227-2019, 2019.

Abbatt, J. P. D., Thomas, J. L., Abrahamsson, K., Boxe, C., Granfors, A., Jones, A. E., King, M. D., Saiz-Lopez, A., Shepson, P. B., Sodeau, J., Toohey, D. W., Toubin, C., von Glasow, R., Wren, S. N., and Yang, X.: Halogen activation via interactions with environmental ice

and snow in the polar lower troposphere and other regions, *Atmos. Chem. Phys*., 12, 6237-6271, https://doi.org/10.5194/acp-12-6237-2012, 2012.

Pöhler, D., et al., *PNAS*, **107**, 6582–6587, doi:10.1073/pnas.0912231107, 2010.

Simpson, WR, Carlson, D, Hönninger, G, Douglas, TA, Sturm, M, Perovich, D and Platt, U.: First-year sea-ice contact predicts bromine monoxide (BrO) levels at Barrow, Alaska better than potential frost flower contact, *Atmos. Chem. Phys*., **7**(3): 621–627, doi:10.5194/acp-7-621-2007, 2007b.

Zhao, X., et al.: A case study of a transported bromine explosion event in the Canadian high arctic, *J. Geophys. Res. Atmos.*, 121, 457-477, 10.1002/2015JD023711, 2016.

---

## Referee Comment (RC2) · Anonymous Referee #3 · 26 Mar 2020

**General comments:**

The manuscript by Seo et al. presents an interesting statistical research work using long-term satellite-based BrO column measurements. The analysis process is comprehensive. The findings, especially the wind-direction based analysis, in this work are important for the atmospheric bromine research community. The only major concern I have is why the author did not use tropospheric BrO column products. I am sure with such valuable 10-yrs observations, the author can provide more important and meaningful results to the research community, if both total and tropospheric BrO columns are used. Otherwise, the manuscript is well written and should be published

after addressing the following comments.

**Specific comments:**

P5 L158: I think the author wants to say all BrO DSCDs (or from which ones?) are fitted by a Gaussian function, and the mode of the function is used in the correction. Anyway, the sentence is not very clear. Please revise it.

P8 L236-237: No information about cloud filtering are provided. BrO enhancement/hotspots induced by large scale low-pressure systems (e.g., the case in Blechschmidt et al., 2016) may accompany with large cloud covers and even precipitations (e.g., Zhao et al., 2017). How these cloudy pixels were treated? What are their impacts (to sensitivies of stratospheric and tropospheric BrO)? I understand some cloudy pixels should be kept for the purpose of this study. But, can you provide statistical analysis with/without cloudy pixels (e.g., with any threshold like cloud fraction < 0.3 or any reasonable one)?

P8 L240-242. It is a very interesting and important figure (Fig. 2). It shows the Canadian archipelagoes are the BrO swamp. But, I am not sure it is misleading or not. I think the author only used the pixels over the sea with sea ice fraction > 5 % (Section 3.1). So, is this selection make any impact over Canadian archipelagoes (i.e., where the land-sea ratio was determined by what?)? Anyway, from Fig. 2, I think this is the region that has larger land to sea ratio, compare to all other studied regions. Please provide more comments and explanations for this important result.

P10 L301-306: The positive surface temperature anomalies coincident within the regions that have low-pressure anomalies. I don't think it is a surprise, but a good indication that these analyses are strongly correlated (warmer surface temperature in the low-pressure system). The factors analyzed in this work (e.g., temperature, wind

speed, and tropopause) are not truly independent in contributing to the enhanced total column BrO. Without separation of the source of bromine enhancement, i.e., enhancement due to dynamic process (low tropopause, more stratospheric bromine) or chemical process (surface bromine explosion) or both, the results presented here are a bit vague and complex. The mechanism of enhanced BrO columns discussed in this and the previous section (Sects. 4.2.1 and 4.2.2) are mostly for surface bromine enhancement (except low tropopause). So, why not performing all the above analyses with satellite tropospheric BrO data? To me, this will provides more insights from this valuable 10-yrs satellite observations.

P10 L294-299: Comparing to the frequency distribution of pressure (i.e., Fig. 3), the results (Fig. 6) here show a significant difference between Arctic and Antarctic. Can the author provide some comments on why we observed such differences? Is this indicate some major differences in the driven factors in total BrO at these two regions? Anyway, similar to my previous comments, the Canadian archipelagos have unique conditions in these analyses (i.e., larger land-sea ratio, thus colder than pure sea ice region in general). With/without this region may affect the frequency distributions.

P10 L317: Well, I thought the community already found the base assumptions supporting frost flower as the direct-source of bromine explosion is over (Abbatt et al., 2012). The surface area of the frost flower is not as large as expected (e.g., Obbard et al., 2009; Roscoe et al., 2011). There are still some hypotheses that frost flower could play some indirect roles in bromine explosion, but please do not say frost flower is a "primary source of bromine explosion events". Otherwise, this will be misleading, and an overlook of all previous research works.

P11 L334-335: I cannot agree with this. The wind speed anomalies in the Canadian archipelagoes are weaker than the other regions mention by the author (e.g., the eastern coast of Greenland). In fact, the wind speed in the Canadian archipelagoes

is lower compare to most of other regions. This is topography determined. The conclusion here is not valid (enhancement of BrO columns related to positive wind anomalies), unless one excludes the Canadian archipelagoes in the frequency analysis (which I would suggest to). Also, even for the high wind regions (the eastern coast of Greenland or centre Arctic sea), I did not see the high frequency of BrO enhancement in Fig. 2. The cause of this might be the high surface wind (10 m wind) is only one of the driven factors for blowing snow induced surface BrO enhancement. But, the author had a discussion of total column frequency (not tropospheric column), which has other major driven factors should be considered. Anyway, perfect separation of all these strongly correlated factors is not possible. But, at least, one can separate the stratospheric signal.

P11-12 L355-363: Fig. 12 is the high wind speeds frequency, which shows that we have more high wind conditions at locations such as Greenland or centre Arctic sea. I agree with this. But, how this can prove high wind speed frequency is consistent with a high frequency of BrO enhancement? I am very confused about this paragraph. For example, if we compare Figs. 2 and 12, we can easily find the eastern coast of Greenland has a low frequency in BrO enhancement but a high frequency in high wind speed. Same for the Canadian archipelagoes, where the high wind is less common but has a very high chance of enhanced BrO columns. I am not challenging the blowing snow scheme, but one should be clear that the transported bromine explosion events may have a different spatial distribution pattern compare to stable shallow boundary layer events. In other words, shallow ones are confined at local, which one might find easy correlation as "low-wind and high BrO" in one place. But, the transported events may be originated or triggered in this 12 m/s wind speed conditions, but transported in relative mild condition (e.g., $< 6$ m/s). Anyway, the analysis done in the next paragraphs is decent and important (L364-416). Wind speed analysis should be done together with wind direction.

P14 L445-447: Since the Canadian archipelago usually has low tropopause, then how this contribute to the BrO hotspot frequency map (i.e., Fig. 2)?

P16 L503-504: As the author already found out, use only wind speed is not sufficient (need to include wind-direction at least). Do you have correlation analysis for different wind directions too? Do we have a better (higher) correlation when we have preferred wind-directions?

P 18: L562-566: These are significant factors that should be addressed before the analysis. I fully understand the limits and difficulties in performing this large scale study (both time and spatial). The paper is well written and meaningful. But, I would suggest the author provide these limits before the beginning of the analysis. The author can inform the reader why the stratospheric correction is not applied (i.e., why not using BrO tropospheric columns).

**Technical corrections:**

P5 L139: Use proper multiple signs in here and thereafter, not letter "x".

P5 L160: Define DSCD.

P28. Fig 1: Use consistent y limits for all four panels (e.g., 1e5). The current selections for each panel are a bit arbitrary.

Figs. 2 and 4: The 0-degree Longitude sign and the 70-degree Latitude sign are jammed.

P13 L421: Please provide the definition of DU (Dobson unit).

P20 L 627: Capitalize each word; change "Geophysical research letters" to "Geophysical Research Letters."

P23 L736: Remove "n/a-n/a".

**Reference**

Abbatt, J. P. D., Thomas, J. L., Abrahamsson, K., Boxe, C., Granfors, A., Jones, A. E., King, M. D., Saiz-Lopez, A., Shepson, P. B., Sodeau, J., Toohey, D. W., Toubin, C., von Glasow, R., Wren, S. N. and Yang, X.: Halogen activation via interactions with environmental ice and snow in the polar lower troposphere and other regions, Atmos. Chem. Phys., 12, 6237–6271, doi:10.5194/acp-12-6237-2012, 2012.

Blechschmidt, A.-M., Richter, A., Burrows, J. P., Kaleschke, L., Strong, K., Theys, N., Weber, M., Zhao, X. and Zien, A.: An exemplary case of a bromine explosion event linked to cyclone development in the Arctic, Atmos. Chem. Phys., 16, 1773–1788, 2016.

Obbard, R. W., Roscoe, H. K., Wolff, E. W. and Atkinson, H. M.: Frost flower surface area and chemistry as a function of salinity and temperature, J. Geophys. Res., 114, doi:10.1029/2009JD012481, 2009.

Roscoe, H. K., Brooks, B., Jackson, A. V., Smith, M. H., Walker, S. J., Obbard, R. W. and Wolff, E. W.: Frost flowers in the laboratory: Growth, characteristics, aerosol, and the underlying sea ice, J Geophys Res-Atmos, 116(D12), doi:10.1029/2010JD015144, 2011.

Zhao, X., Weaver, D., Bognar, K., Manney, G., Millán, L., Yang, X., Eloranta, E., Schneider, M. and Strong, K.: Cyclone-induced surface ozone and HDO depletion in

the Arctic, Atmos. Chem. Phys., 17(24), 14955–14974, doi:10.5194/acp-17-14955-2017, 2017.

---

## Author Comment (AC1) · 21 Jul 2020

**Response to anonymous referee #2**

We thank the reviewer for his/her constructive comments and suggestions, which helped us to improve our manuscript. We have addressed the questions as follows:

**General comments:**

This manuscript reports a statistic analysis of polar total column BrO and its corresponding meteorological parameters (including temperature, surface pressure, wind seed, direction and tropopause height), using 10-year GOME-2 BrO and meteorological data. A further Spearman rank correlation analysis was applied to assess the dependence of total column BrO on various meteorological parameters. Some interesting results regarding the spatial distribution of enhanced BrO and its relation to the above meteorological parameters are reported which is welcome. The authors also attempted to link the relationship to relevant physical/chemical mechanism(s) being proposed previously. This study indeed adds some new knowledge to help a better understanding of the underlying processes involved. However, I found their attempt to evaluate some relevant mechanisms is dangerous and inappropriate. This is because the bromine source is a complex function of multiple (rather than one) factors and the relations between them are largely non-linear, rather than linear. For example, the SSA production from blowing snow is a complex function of wind speed, temperature and relative humidity (Yang et al., 2019), more specifically, the SSA production (and bromine flux) is actually proportional to the sublimation flux of blowing snow, rather than individual parameters (winds, temperature and RH). Secondly, these meteorological parameters examined are not independent, instead they are deeply correlated (also pointed by the authors). For example, they are deeply correlated during a storm system. For this reason, I suggest the authors add a table to show the cross-correlation among these meteorological parameters during enhanced BrO cases and discuss their implications to deriving your conclusions. Thirdly, it is not correct to assume that a higher correlation coefficient between two factors than others necessary means this link is stronger than other links. Unfortunately, this assumption seems being used in this manuscript to make some conclusions. For the above reasons, I suggest a major revision before recommend it to publish in ACP.

We agree with the reviewer's comments that the mechanisms associated with bromine release are a function of multiple factors and that the meteorological factors examined in this study are not independent but correlated. To provide additional information on this, cross-correlations between the relevant meteorological parameters during the enhancement of total BrO VCD have been investigated and analyzed. The results were added in section 4.3 of the revised manuscript as suggested. Also, time-lagging effects on the relationship between meteorological parameters and total BrO VCD have been investigated. These additional test results and their analysis are

described in section 4.3 (P16 L514) as follows (blue text):

[revised manuscript text omitted]

**Specific comments:**

P2L35-36: 'Overall, the BrOx catalytic destruction cycles are significant but not as important as those of ClOx', though this statement is correct, recent modelling-based work indicates the cross-halogen reactions (e.g. between Br and Cl) are important in terms of stratospheric ozone depletion, see recent work by Fraiser et al. (2019).

We have revised the text and added a sentence as suggested (blue text) in section 1 (P2 L34):

"Inorganic bromine is then involved in a variety of stratospheric chemical reactions. The ozone catalytic removal cycles involving bromine atoms, Br, and bromine monoxide (BrO), known as BrOx, have higher chain lengths than those involving chlorine atoms, and chlorine monoxide (ClO), together known as ClOx. Although bromine is more effective at depleting ozone, it cannot be said that BrOx catalytic destruction cycles are more important than those of ClOx since bromine released by man is much less than chlorine. Also, recent studies found that the sensitivity of the ozone depletion to bromine concentration depends on the amount of chlorine present by the chlorine/bromine cross reaction (Yang et al., 2014; Dennison et al., 2019). More details of the current assessment of the role of bromine in the stratosphere are reported in the recent ozone assessment (WMO, 2018)."

Yang, X., Abraham, N. L., Archibald, A. T., Braesicke, P., Keeble, J., Telford, P. J., Warwick, N. J., and Pyle, J. A.: How sensitive is the recovery of stratospheric ozone to changes in concentrations of very short-lived bromocarbons?, Atmos. Chem. Phys., 14, 10431–10438, https://doi.org/10.5194/acp-14-10431-2014, 2014.

Dennison, F., Keeble, J., Morgenstern, O., Zeng, G., Abraham, N. L., and Yang, X.: Improvements to stratospheric chemistry scheme in the UM-UKCA (v10.7) model: solar cycle and heterogeneous reactions, Geosci. Model Dev., 12, 1227–1239, https://doi.org/10.5194/gmd-12-1227-2019, 2019.

P2L43-44: 'An important tropospheric inorganic bromine source is the polar sea ice region (Kaleschke et al., 2004 and references therein)'. It would be better to cite a review paper (e.g. Abbatt et al., 2012) here, as Kaleschke et al.'s paper specifically addressed the importance of frost flowers. Or you can specify various proposed candidates of sea ice sourced bromine from frost flowers (Kaleschke et al. 2004), first-year sea-ice (Simpson et al., 2007b; Pöhler, et al., 2010), sea salt aerosol produced from blowing snow (Yang et al., 2008), stratospheric origin (Salawitch et al., 2010), snowpack photochemistry (e.g. Pratt et al., 2013) and sea spray from open leads (e.g. Peterson et al., 2015). See references shown below.

As suggested, we have revised the text in section 1 (P2 L43) and added references as:

"An important tropospheric bromine source is the polar sea ice region (Abbatt et al., 2012 and references therein). First-year sea-ice (Simpson et al., 2007b; Pöhler, et al., 2010), frost flowers (Kaleschke et al. 2004), sea salt aerosol produced from blowing snow (Yang et al., 2008), snowpack photochemistry (Yang et al., 2008; Pratt et al., 2013), stratospheric origin (Salawitch et al., 2010), and sea spray from open leads (Peterson et al., 2015) have been proposed as major sources of bromine in polar region."

Abbatt, J. P. D., Thomas, J. L., Abrahamsson, K., Boxe, C., Granfors, A., Jones, A. E., King, M.

D., Saiz-Lopez, A., Shepson, P. B., Sodeau, J., Toohey, D. W., Toubin, C., von Glasow, R., Wren, S. N., and Yang, X.: Halogen activation via interactions with environmental ice and snow in the polar lower troposphere and other regions, Atmos. Chem. Phys., 12, 6237–6271, https://doi.org/10.5194/acp-12-6237-2012, 2012.

Simpson, W. R., Carlson, D., Hönninger, G., Douglas, T. A., Sturm, M., Perovich, D., and Platt, U.: First-year sea-ice contact predicts bromine monoxide (BrO) levels at Barrow, Alaska better than potential frost flower contact, Atmos. Chem. Phys., 7, 621–627, https://doi.org/10.5194/acp-7-621-2007, 2007b.

Pöhler, D., Vogel, L., Frieß, U., and Platt, U.: Observation of halogen species in the Amundsen Gulf, Arctic, by active long-path differential optical absorption spectroscopy, P. Natl Acad. Sci., 107, 6582–6587, doi:10.1073/pnas.0912231107, 2010.

Pratt, K. A., Custard, K. D., Shepson, P. B., Thomas, D. A., Pohler, D., General, S., Zielcke, J., Simpson, W. R., Platt, U., Tanner, D. J., Huey, L. G., Carlson, M., and Stirm, B. H.: Photochemical production of molecular bromine in arctic surface snowpacks, Nat. Geosci., 6, 351–356, doi:10.1038/ngeo1779, 2013

P4L119-123: 'In this study, to overcome this snapshot treatment of elevated BrO events during polar spring, and to obtain a more general understanding of the enhancements of total BrO columns, we …' I do not agree that snapshot study is less-advantaged comparing to general statistic study. They both have their individual advantage and disadvantage, they may compensate each other at some point. I suggest a slight change to the tone used in the statement.

As suggested, we changed the tone used in the statement as follows (blue text) in section 1 (P4 L119):

"As mentioned above, many studies on the possible sources of BrO enhancements and the driving meteorological conditions in polar regions have been conducted using ground-based and satellite measurements. Study results clearly indicate that meteorological conditions affect the processes of BrO enhancements in several ways. These previous studies mainly focus on specific case-studies or analysis using relatively short-term datasets. This study aims at adding to this body of knowledge and to obtain a more general and comprehensive understanding of the enhancements of total BrO columns using a consistent long-term dataset. Therefore, we statistically analyse the spatial distribution of occurrence frequency of enhanced total BrO column and its relationship to various meteorological parameters in the Arctic and Antarctic sea ice regions by using a 10 year long-term dataset. In particular, the relationship between total BrO vertical columns retrieved from GOME-2A/2B and meteorological fields including sea level pressure, surface level wind speed and direction, surface air temperature, and tropopause height were investigated."

P15/P18: in P15L476-477 'Surface wind speed is not significantly correlated with total BrO vertical column density in either the Arctic or Antarctic.' A similar statement is also in the summary (P18L557). However, they are contrary to the results shown in Table 3 and Fig. 20, where it shows 'Note that all Spearman's rank correlations are significant (p<0.001)'. Is surface wind speed significantly correlated with total BrO vertical column density or not?

In a statistical hypothesis test, p-value and r are indicators of different statistical meanings, finding something that is likely above chance (p-value) or finding meaningful correlations between variables (r). Therefore, having a low correlation coefficient and significant p-value between the surface wind speed and total BrO vertical column density is not a contradictory concept.

To avoid misunderstanding in the interpretation of the analysis results, the sentences (P15 L476-477 and P18 L557) has been changed as follows in the revised manuscript (blue text):

"Surface wind speed has correlation coefficients close to 0 with total BrO vertical column density in both the Arctic and Antarctic, and even for wind speeds above 8 m·s⁻¹, which can cause blowing snow, there is no clear relationship between them." This could either be due to stratospheric air dominating the total BrO column or due to the large variability of wind speed conditions under which tropospheric BrO explosion events occur according to previous studies described above.

"Sea level pressure and surface level wind speed are negatively and positively correlated with the total BrO vertical column density, respectively, but their correlation coefficients are low and the strengths of the relationships are weak."

P17L533-4 and L552-555, the two statements are actually duplicated, either delete one or move one to section 4.3.2.

As suggested, we have revised sentences (P17 L553-555) as follows in the revised manuscript:

"One remarkable point is that the temperature is negatively correlated with total BrO column in most sea ice regions, but has a positive correlation over the central Arctic, which is the same as the result of the temperature anomaly pattern discussed above. "

P27 Table 3: given that the occurrence of blowing snow has a threshold wind speed, e.g. 7~8 m/s, I suggest add an extra row in the Table to show the correlation coefficients for wind speeds > 8 m/s (or >12 m/s).

We have added an extra row in Table 3 for the Spearman rank correlation coefficients of wind speeds > 8m/s as suggested, and revised the text as in section 4.3 (P15 L476-477):

"Surface wind speed has correlation coefficients close to 0 with the total BrO vertical column density in both the Arctic and Antarctic, and even for wind speeds above 8 m·s$^{-1}$, which can cause blowing snow, there is no clear relationship between them."

**Table 3.** Spearman rank correlation coefficients between total BrO VCDs and four meteorological parameters. The results are shown separately for different months in spring in the Arctic and Antarctic. Note that all Spearman's rank correlation coefficients are significant (p < 0.001). Abbreviation: MA (March to April), SO (September to October).

| | Arctic | | | Antarctic | | |
|---|---|---|---|---|---|---|
| | **Mar** | **Apr** | **MA** | **Sep** | **Oct** | **SO** |
| **Sea level pressure** | -0.032 | -0.098 | -0.070 | -0.228 | -0.053 | -0.130 |
| **Temperature** | -0.242 | -0.149 | -0.180 | -0.305 | -0.110 | -0.193 |
| **Wind speed** | -0.011 | 0.067 | 0.035 | 0.038 | 0.043 | 0.040 |
| (v ≥ 8 m·s$^{-1}$) | -0.002 | 0.026 | 0.010 | 0.041 | 0.035 | 0.037 |
| **Tropopause height** | -0.336 | -0.298 | -0.315 | -0.400 | -0.307 | -0.345 |

**Minors:**

P1L26: change 'they release bromine atoms and Br and bromine monoxide (BrO)...' to 'they release bromine atoms, e.g. Br and bromine monoxide (BrO) ...'

This has been corrected in the revised manuscript.

P1L28: Theys et al., 2009b should be 2009a, as Theys et al., 2009a has not been cited.

Thanks for pointing this out. We have corrected all relevant references.

P2L31: change '2Br2' to 'Br2'

This has been corrected in the revised manuscript.

P2L32: again you cited 'They et al., 2009b' before their 2009a paper.

This has been corrected in the revised manuscript. Theys et al., 2009a has been cited before in P1L28.

P3L67: Theys et al., 2009a should be cited before 2009b.

This has been corrected in the revised manuscript.

P4L112: 'A bromine explosion event linked to cyclone development in the Arctic was investigated by Blechshmidt et al. (2016).' The same event was also studied by Zhao et al., 2016.

As suggested, we have added the reference Zhao et al., 2016 in the revised manuscript.

"A bromine explosion event linked to cyclone development in the Arctic was investigated by Zhao et al. (2015) and Blechschmidt et al. (2016)."

Zhao, X., Strong, K., Adams, C., Schofield, R., Yang, X., Richter, A., Frieß, U., Blechschmidt, A.-M., and Koo, J.-H.: A case study of a transported bromine explosion event in the Canadian high Arctic, J. Geophys. Res. Atmos., 121, 457–477, doi:10.1002/2015JD023711, 2016.

---

## Author Comment (AC2) · 21 Jul 2020

**Response to anonymous referee #3**

We thank the reviewer for his/her detailed and constructive comments and suggestions, which helped us to improve our manuscript. We have addressed the questions as follows:

**General comments:**

The manuscript by Seo et al. presents an interesting statistical research work using long-term satellite-based BrO column measurements. The analysis process is comprehensive. The findings, especially the wind-direction based analysis, in this work are important for the atmospheric bromine research community. The only major concern I have is why the author did not use tropospheric BrO column products. I am sure with such valuable 10-yrs observations, the author can provide more important and meaningful results to the research community, if both total and tropospheric BrO columns are used. Otherwise, the manuscript is well written and should be published after addressing the following comments.

1) P5 L158: I think the author wants to say all BrO DSCDs (or from which ones?) are fitted by a Gaussian function, and the mode of the function is used in the correction. Anyway, the sentence is not very clear. Please revise it.

   As suggested, we have revised sentences (blue text) in P5 L153-160 as follows:

When using a Pacific background spectrum, the retrieved differential slant columns (DSCD) need to be corrected by adding the BrO slant column over that region. Here we follow earlier studies (Richter et al., 2002; Sihler et al., 2012; Seo et al., 2019) and assume a BrO vertical column of $V_{norm,ref} = 3.5 \times 10^{13}$ molec cm$^{-2}$ over the Pacific reference sector. The corresponding BrO SCD is computed by multiplying the $V_{norm,ref}$ with the geometric air mass factor $A_{geo}$. $A_{geo}$ is defined as:

$$A_{geo} = \frac{1}{\cos(SZA)} + \frac{1}{\cos(VZA)} \qquad \text{(SZA: solar zenith angle, VZA: viewing zenith angle)} \quad (1)$$

As the differential BrO slant columns (DSCD) over the Pacific are not exactly 0, the mode μ of a Gaussian fitted to their distribution is taken into account for the normalization correction:

$$SCD = DSCD + A_{geo} \cdot V_{norm,ref} - \mu \qquad (2)$$

2) P8 L236-237: No information about cloud filtering are provided. BrO enhance-ment/hotspots induced by large scale low-pressure systems (e.g., the case in

Blechschmidt et al., 2016) may accompany with large cloud covers and even precipitations (e.g., Zhao et al., 2017). How these cloudy pixels were treated? What are their impacts (to sensitivities of stratospheric and tropospheric BrO)? I understand some cloudy pixels should be kept for the purpose of this study. But, can you provide statistical analysis with/without cloudy pixels (e.g., with any threshold like cloud fraction < 0.3 or any reasonable one)?

No cloud filtering was applied in this study on the enhancement of BrO in polar regions. The reasons for this decision and potential errors related to this exclusion of cloud filtering have been added to the revised manuscript as follows (blue text):

In section 4.1 (P8 L237):

In this study, a reference grid with 200×200 km resolution was used. The enhanced BrO occurrence frequency $f_{EBrO}$ was calculated by dividing the number of pixels classified as enhanced BrO column in a given reference grid cell by the total number of satellite pixels within the reference grid cell. "One thing to note is that cloud filtering was not applied in this study. Clouds can affect the BrO column retrieval generally in three ways: (1) the albedo effect related to the increase of the reflectivity and sensitivity for cloudy scenes, (2) the enhancement of optical light path due to multiple scattering inside clouds, and (3) the shielding effect that hides trace gases below clouds. The first two effects could increase the absorption of trace gases and lead to values greater than the actual total BrO column, while the third effect leads to an underestimation of the total BrO column (Antón and Loyola, 2011). Therefore, as clouds affect the BrO retrieval, it is necessary to consider the presence and characteristics of clouds for accurate BrO analysis. However, obtaining long-term reliable cloud products such as cloud fraction and cloud top height over the polar sea ice regions is difficult since detecting clouds and retrieving their properties over a bright snow/sea ice surface from satellite measurements are difficult (Heidinger and Stephens, 2000). Inaccurate cloud data may cause errors in statistical analysis using long-term data. Also, the difference between cloud free and cloudy conditions in polar sea ice regions is relatively small due to the bright surface (Figure 1 in Blechschmidt et al., 2016). Based on these considerations, this study did not attempt to correct the effects of clouds on the enhancement of BrO columns."

Antón, M. and Loyola, D.: Influence of cloud properties on satellite total ozone observations, J. Geophys. Res., 116, D03208, doi:10.1029/2010JD014780, 2011.

Heidinger, A. K. and Stephens, G. L.: Molecular Line Absorption in a Scattering Atmosphere. Part II: Application to Remote Sensing in the O2 A band, J. Atmos. Sci., 57, 1615–1634, doi:10.1175/1520-0469(2000)057<1615:MLAIAS>2.0.CO;2, 2000

3) P8 L240-242. It is a very interesting and important figure (Fig. 2). It shows the Canadian archipelagoes are the BrO swamp. But, I am not sure it is misleading or not. I think the author only used the pixels over the sea with sea ice fraction > 5 % (Section 3.1). So, is this selection make any impact over Canadian archipelagoes (i.e., where the land-sea ratio was determined by what?)? Anyway, from Fig. 2, I think this is the region that has larger land to sea ratio, compare to all other studied regions. Please provide more comments and explanations for this important result.

[Figure]

Figure A2. Monthly spatial distribution of the number of (a) all data collected (sea ice concentration > 5%) and (b) selected data for enhanced BrO cases over the Arctic during the study period.

The reviewer is right that the limitation to measurements over sea-ice impacts on the statistics in the Canadian archipelago. As shown in Figure A2, due to the large land-sea ratio in the Canadian archipelago, the number of data points collected for the analysis is low compared to other regions. However, since enhanced BrO cases are observed more frequently than in other regions, the relative occurrence frequency of enhanced total BrO column is high in this region. Since a relatively small number of data points was used in the statistical analysis compared to other regions, the monthly histograms of the total BrO VCD (Fig. 1), the frequency distribution of meteorological factors for the mean field and enhanced BrO cases (i.e. Fig. 3, 6, 9, and 17) as well as Spearman rank correlation analysis (Table 3) will not change significantly if the Canadian archipelago is included or excluded from the study domain.

We have added comments in Section 4.1 (P8 L241) as follows:

In the Arctic, enhanced total BrO columns are frequently observed over the north of the Canadian coast with a frequency of ~0.25 (25 %). Also, over the Hudson Bay, fEBrO is higher than ~0.18 in both March and April. "Due to the large land-sea ratio in the Canadian archipelago, the number of data collected for the analysis is low compared to other regions. However, since enhanced BrO cases are observed more frequently than in other regions, the relative occurrence frequency of enhanced total BrO column is high in this region." The spatial distribution patterns of $f_{EBrO}$ are mostly similar in March and April, but relatively high values of ~0.15 are observed over the Chukchi Sea and the East Siberian Sea in April.

4) P10 L301-306: The positive surface temperature anomalies coincident within the regions that have low-pressure anomalies. I don't think it is a surprise, but a good indication that these analyses are strongly correlated (warmer surface temperature in the low-pressure system). The factors analyzed in this work (e.g., temperature, wind speed, and tropopause) are not truly independent in contributing to the enhanced total column BrO. Without separation of the source of bromine enhancement, i.e., enhancement due to dynamic process (low tropopause, more stratospheric bromine) or chemical process (surface bromine explosion) or both, the results presented here are a bit vague and complex. The mechanism of enhanced BrO columns discussed in this and the previous section (Sects. 4.2.1 and 4.2.2) are mostly for surface bromine enhancement (except low tropopause). So, why not performing all the above analyses with satellite tropospheric BrO data? To me, this will provides more insights from this valuable 10-yrs satellite observations.

We agree with your comment that the factors analyzed in this study are not truly independent. In this regard, additional analysis was performed on the cross correlation between meteorological parameters during the enhancement of total BrO columns and added in section 4.3 (P16 L514) of the revised manuscript as follows:

"The relationship between individual meteorological parameters and the total BrO vertical column investigated above illustrates how each meteorological parameter is linked to BrO variations in terms of temporal and spatial distribution. However, since meteorological parameters are not independent of each other and vary systematically in general, cross-relationships between meteorological parameters affecting directly or indirectly BrO variations should also be considered. For example, Yang et al. (2019) showed that the sea salt aerosol (SSA) production affecting the enhancement of BrO at the tropospheric level is proportional to the sublimation flux of blowing snow which is affected by various meteorological parameters including surface wind speed, temperature and relative humidity. Also, Zhao et al. (2015) and Blechschmidt et al. (2016) showed that large-scale enhanced BrO plumes over the Beaufort Sea are associated with weather systems which change the various relevant meteorological parameters together. They also demonstrated that the size and lifetime of BrO plumes depend on the development stage of the weather system. Therefore, cross-correlations between meteorological parameters for those data having enhanced total BrO columns were investigated (see Table 4). During the occurrence of enhanced total BrO, sea level pressure has a negative correlation with surface level temperature and wind speed, while it has a positive correlation with tropopause height. For example, the development of a low pressure system during the enhancement of BrO columns may correlate with a decrease in tropopause height as well as an increases in surface level air temperature and wind speed. Although the correlation coefficients found are not large, results show that sea level pressure is linked with both surface level meteorological conditions and the tropopause height which can account for stratospheric dynamics. Indeed, pressure systems which usually evolve due to interactions of temperature differences in the atmosphere derive directly the airflow motion within the troposphere and also may affect the tropopause height in relation to the convergence or divergence of air masses. It is also interesting to note from Table 4 that the tropopause height has insignificant correlations with surface level meteorological parameters during the BrO enhancements, except for the air temperature in the Arctic, which is predictable since the tropopause height is a factor more closely related to stratospheric dynamics compared to the surface level weather system.

Table 4. Cross-correlations of meteorological parameters for the enhanced total BrO cases in the Arctic and Antarctic sea ice region.

| | Arctic | | | | Antarctic | | | |
|---|---|---|---|---|---|---|---|---|
| | Sea level pressure | Temperature | Wind speed | Tropopause height | Sea level pressure | Temperature | Wind speed | Tropopause height |
| Sea level pressure | 1 | -0.186 | -0.228 | 0.186 | 1 | -0.163 | -0.195 | 0.173 |
| Temperature | | 1 | 0.183 | 0.249 | | 1 | 0.158 | -0.009 |
| Wind speed | | | 1 | -0.013 | | | 1 | -0.05 |
| Tropopause height | | | | 1 | | | | 1 |

Also, as the reviewer points out, tropopause height is related to the stratospheric contribution, while surface air temperature and wind are related to the tropospheric contribution on the BrO column.

We agree that in principle, it is more desirable to perform the analysis on tropospheric BrO vertical columns to more clearly distinguish the effect of meteorological factors on the stratospheric and tropospheric BrO columns. However, there are two main reasons for using total BrO vertical column instead of tropospheric BrO vertical column in this study.

First, meteorological systems contribute to both the troposphere and stratosphere as changes in atmospheric pressure, tropopause height, temperature, wind speed and direction interact with each other. For example, high or low pressure systems evolve due to atmospheric temperature differences and the pressure systems may affect the tropopause height and surface level winds. Therefore, this study aims to investigate how the meteorological systems generally affect the total BrO column, rather than attempting to separate the effects on the enhancement of BrO in the troposphere and stratosphere.

The second reason is that the accuracy of tropospheric BrO retrieved by applying a stratospheric correction and tropospheric AMF to satellite observation data over a long period of 10 years is difficult to assess. One of the most used methods for stratospheric correction is the climatological approach developed by Theys et al. (2009), a method using stratospheric BrO profiles based on a parametrization using 3D chemistry transport model BASCOE data. The advantage of this stratospheric correction method is the reflection of both dynamical and chemical effects on the stratosphere using the intermediate input data of total $O_3$ and stratospheric $NO_2$ columns from satellite measurements which are rather easily accessible compared to other model data. However, the BASCOE model climatology look-up-table (LUT) developed in Theys et al. (2009) is a result using three years of data (from April 2003 to March 2006) of a low resolution (3.75° x 5°) model run. Since this study is based on the analysis of long-term data for 2008-2018, correction factors for the long-term trend would have to be considered. Also, dynamical effects are not necessarily well reflected in the climatology because of the low spatial resolution of the model run, resulting in large uncertainties of tropospheric columns in situations with large tropopause changes. We therefore decided to use total columns for this study. Once well validated separation methods are available, the analysis should be repeated on tropospheric columns.

The reasons for using total BrO columns to investigate the enhancement of BrO and its relation to meteorological factors have been added to the revised manuscript.

- In section 1 (P4 L126):

In particular, the relationship between total BrO vertical columns retrieved from GOME-2A/2B and meteorological fields including sea level pressure, surface level wind speed

and direction, surface air temperature, and tropopause height were investigated. "The reason for using total BrO columns instead of tropospheric and stratospheric columns separately to examine the relationship with the meteorological fields is that existing separation methods for satellite BrO data are difficult to apply to a long-term dataset in both hemispheres. They also have large uncertainties in connection with low pressure systems and large tropopause height changes which affect both stratospheric and tropospheric columns. This study aims to investigate how the meteorological system generally affects the total BrO column, rather than separate the effects on the enhancement of BrO in each atmospheric layer." Differences in meteorological conditions and their regional characteristics between high BrO situations and the mean field were investigated in order to better understand meteorological effects on processes involved in BrO enhancements. Finally, based on Spearman rank correlation analysis, the degree of influence of different meteorological parameters on total BrO columns was evaluated and the most important meteorological parameters, influencing BrO, and their regional patterns were identified.

5) P10 L294-299: Comparing to the frequency distribution of pressure (i.e., Fig. 3), the results (Fig. 6) here show a significant difference between Arctic and Antarctic. Can the author provide some comments on why we observed such differences? Is this indicate some major differences in the driven factors in total BrO at these two regions? Anyway, similar to my previous comments, the Canadian archipelagos have unique conditions in these analyses (i.e., larger land-sea ratio, thus colder than pure sea ice region in general). With/without this region may affect the frequency distributions.

The pattern in the frequency distribution of sea level pressure between the enhanced BrO cases and the mean field is similar in both the Arctic and Antarctic, whereas the difference in the frequency distribution of the surface air temperature is large between the Arctic and Antarctic. The reason why the difference in the frequency distribution of the surface air temperature is more apparent in the Antarctic than the Arctic can be identified in the Fig. 7 and 8. The spatial distribution of surface air temperature anomalies between the enhanced BrO and the mean field shows a slight air temperature increase over the central Arctic. The detection of enhanced BrO in the central Arctic (80-90 °N) may be related to the transport of plumes with high BrO occurring at relatively lower latitudes (70-80 °N). The Canadian archipelago region with a larger land-sea ratio is one of the coldest regions in the Arctic as much as the central Arctic, but the temperature anomalies map shows that the temperature is lower when BrO enhancements occur (see Fig. 7). Thus, it is expected that the pattern will not change largely if the Canadian archipelago is not included in the statistical analysis since the main cause of the small difference of the frequency distributions of air temperature between the enhanced BrO cases and the mean field in the Arctic is attributed to the temperature rise in the central Arctic.

Unlike the Arctic, the difference in the range of air temperature frequency distribution between the enhanced BrO cases and the mean field in the Antarctic is large. In the Antarctic, it can be seen that in most regions except Antarctic coastal regions, the surface air temperature decreases strongly when total BrO enhancements occur, especially in the sea ice margin (55-60 $^o$S). This is related to the contrast between relatively warm air coming from the open ocean region and cold air from the continent / sea ice region which is the result of the very different sea-land distribution in the two hemispheres.

6) P10 L317: Well, I thought the community already found the base assumptions supporting frost flower as the direct-source of bromine explosion is over (Abbatt et al., 2012). The surface area of the frost flower is not as large as expected (e.g., Obbard et al., 2009; Roscoe et al., 2011). There are still some hypotheses that frost flower could play some indirect roles in bromine explosion, but please do not say frost flower is a "primary source of bromine explosion events". Otherwise, this will be misleading, and an overlook of all previous research works.

We agree that the sentence was misleading and revised it as following:

"Frost flowers which are water ice, coated with brine can act as a  source of bromine explosion events."

7) P11 L334-335: I cannot agree with this. The wind speed anomalies in the Canadian archipelagoes are weaker than the other regions mentioned by the author (e.g., the eastern coast of Greenland). In fact, the wind speed in the Canadian archipelagoes is lower compare to most of other regions. This is topography determined. The conclusion here is not valid (enhancement of BrO columns related to positive wind anomalies), unless one excludes the Canadian archipelagoes in the frequency analysis (which I would suggest to).

Also, even for the high wind regions (the eastern coast of Greenland or centre Arctic sea), I did not see the high frequency of BrO enhancement in Fig. 2. The cause of this might be the high surface wind (10 m wind) is only one of the driven factors for blowing snow induced surface BrO enhancement. But, the author had a discussion of total column frequency (not tropospheric column), which has other major driven factors should be considered. Anyway, perfect separation of all these strongly correlated factors is not possible. But, at least, one can separate the stratospheric signal.

As suggested, we have added descriptions (blue text) for the Canadian archipelago which has different characteristics from most other Arctic regions with respect to the relationship between the surface wind speed and the total BrO column enhancement in the revised manuscript.

- In section 4.2.3 (P11 L326-335):

"Next, surface level wind speed is investigated to evaluate how this may affect the occurrence of total BrO column enhancements. Figure 9 shows the frequency distribution of wind speed at 10 m for the average field and for enhanced BrO cases of the 10 years of measurements in the Arctic and Antarctic sea ice region. The distribution is shifted towards high wind speeds in both polar regions for enhanced total BrO vertical columns, the increase in wind speed being more pronounced in the Antarctic region. The difference in wind speeds is also confirmed by the spatial distribution maps (Fig. 10 and 11). Higher wind speeds are observed in most Arctic and Antarctic regions for situations with enhanced total BrO columns compared to the mean field. In particular, differences in wind speed of more than 5 $m \cdot s^{-1}$ and high wind speeds of over 10 $m \cdot s^{-1}$ are found at specific regions of the Arctic such as the eastern coast of Greenland, the Bering Strait and the central Arctic. However, the Canadian archipelago has weaker wind speed anomalies with a low wind range of 3-5 $m \cdot s^{-1}$ in both the mean field and enhanced BrO situations due to the effect of topography. In the Antarctic region, wind speeds greater than 12 $m \cdot s^{-1}$ are predominantly observed over the sea ice margins and some of the Antarctic coastline. In the Antarctic region, wind speeds greater than 12 $m \cdot s^{-1}$ are predominantly observed over the sea ice margins and some of the Antarctic coastline. Our results show that enhancements of BrO columns are mainly related to positive wind speed anomalies except for some areas such as the Canadian archipelago. The enhanced BrO columns in the Canadian archipelago, an area where total BrO hotspots are frequently detected with low surface wind speeds, may be attributed to local production/recycling of reactive bromine under a stable boundary layer or an increase in stratospheric BrO."

We partly agree with the statement that separating troposphere and stratosphere would be simplifying the interpretation of the results. However, as discussed above (Question #4), we feel that the accuracy of current separation methods would introduce additional uncertainty. Our results of the relationship between total BrO vertical columns and surface wind speeds show large positive wind speed anomalies (Fig. 10) and high values of the relative frequency of strong surface winds during the occurrence of enhanced total BrO (Fig. 12) along the eastern coast of Greenland and the central Arctic. From these results, we can suppose that although the occurrences of enhanced total BrO columns in these regions are not as high as in other regions, the supply of reactive bromine sources from blowing snow events caused by high surface wind speeds and the tropospheric bromine explosion mechanism contribute to the increase in total BrO columns.

8) P11-12 L355-363: Fig. 12 is the high wind speeds frequency, which shows that we have more high wind conditions at locations such as Greenland or centre Arctic sea. I agree

with this. But, how this can prove high wind speed frequency is consistent with a high frequency of BrO enhancement? I am very confused about this paragraph. For example, if we compare Figs. 2 and 12, we can easily find the eastern coast of Greenland has a low frequency in BrO enhancement but a high frequency in high wind speed. Same for the Canadian archipelagoes, where the high wind is less common but has a very high chance of enhanced BrO columns. I am not challenging the blowing snow scheme, but one should be clear that the transported bromine explosion events may have a different spatial distribution pattern compare to stable shallow boundary layer events. In other words, shallow ones are confined at local, which one might find easy correlation as "low-wind and high BrO" in one place. But, the transported events may be originated or triggered in this 12 m/s wind speed conditions, but transported in relative mild condition (e.g., < 6 m/s). Anyway, the analysis done in the next paragraphs is decent and important (L364-416). Wind speed analysis should be done together with wind direction.

We agree with the reviewer that local BrO enhancements and transported BrO events need to be treated separately, and that transported events are more difficult to link to the driving mechanisms. We also would like to point out that Figure 12 shows regions, where high wind speeds are found frequently during enhanced BrO events, which we take as indication that the high wind speed could be related to the formation of BrO. This is in line with Figures 10 and 11 showing regions where the probability for high wind speeds increases in the presence of BrO.

To clarify the conclusions that can be inferred from the occurrence frequency map of enhanced total BrO column (Fig. 2) and the relative frequency map of high wind speeds during enhanced total BrO occurrences (Fig. 12), the following blue text has been added and revised in section 4.2.3:

- In section 4.2.3 (P12 L363):

The spatial distribution map of surface wind speed anomalies derived in this study shows that during the enhancement of total BrO vertical columns, wind speeds are generally enhanced. However, the average wind speed field during the high BrO cases shows values of 6-8 m·s-1 in most areas. For the tropospheric bromine explosion events created by strong winds, previous studies indicate that wind speeds above ~12 m·s$^{-1}$ are required. The regions that satisfy this wind speed threshold consistently are confined to the Bering Strait, the central Arctic, and the east coast of Greenland in the Arctic and the Antarctic sea ice margins and some coastal locations. This behaviour is clearly identified in the spatial distribution maps of the relative frequency of high wind speeds for the occurrence of enhanced total BrO columns (see Fig. 12) which show where strong surface winds contribute to the enhancement of BrO columns. In Fig. 12, high frequencies above 30 % are found over the central Arctic and eastern coast of Greenland in the Arctic, whereas in the Antarctic, they are most frequently detected around the marginal ice zone of the Weddell and Ross Sea.

 "In particular, although the central Arctic and eastern coast of Greenland are regions where enhanced total BrO columns are not frequently detected in the Arctic as shown in Fig. 2, it is clear that the occurrence of enhanced total BrO columns in the corresponding regions is often associated with higher wind speeds (Fig. 10 and 12). This could indicate that tropospheric bromine sources from blowing snow events generated by high wind speeds are important in these areas."

- In section 4.2.3 (P11 L335):

"Our results show that enhancements of BrO columns are mainly related to positive wind speed anomalies except for some areas such as the Canadian archipelago. The enhanced BrO columns in the Canadian archipelago, an area where total BrO hotspots are frequently detected with low surface wind speeds, may be attributed to local production/recycling of reactive bromine under a stable boundary layer or an increase in stratospheric BrO."

10) P16 L503-504: As the author already found out, use only wind speed is not sufficient (need to include wind-direction at least). Do you have correlation analysis for different wind directions too? Do we have a better (higher) correlation when we have preferred wind-directions?

The reason why the correlation analysis between the surface wind direction and total BrO vertical column was not performed in Section 4.3 is that wind direction is a categorical variable (8 wind direction groups) unlike other meteorological parameters such as sea level pressure, air temperature, wind speed and tropopause height. Instead of the correlation analysis between the wind direction (categorical variable) and total BrO VCD (numerical variable), to investigate the dominant wind direction related to the enhancement of total BrO columns, we examined the spatial distribution of relative frequency for each wind direction during the occurrence of enhanced total BrO columns (Fig. 13-16 in Section 4.2.3). Although the statistical analysis whether preferred wind directions have higher correlation coefficients with total BrO columns has not been performed, it can be assumed by spatial distribution maps of relative frequency anomalies of wind direction between enhanced total BrO cases and the mean field (Fig. 14 and 16). Positive frequency anomalies mean that the corresponding wind direction has a higher correlation with total BrO column than other wind directions in the study area.

11) P18 L562-566: These are significant factors that should be addressed before the analysis. I fully understand the limits and difficulties in performing this large scale study

(both time and spatial). The paper is well written and meaningful. But, I would suggest the author provide these limits before the beginning of the analysis. The author can inform the reader why the stratospheric correction is not applied (i.e., why not using BrO tropospheric columns).

As suggested, we have added limitations and potential weaknesses of this study in the revised manuscript as follows:

- In section 1 (P4 L126):

In particular, the relationship between total BrO vertical columns retrieved from GOME-2A/2B and meteorological fields including sea level pressure, surface level wind speed and direction, surface air temperature, and tropopause height were investigated. "The reason for using total BrO columns instead of tropospheric and stratospheric columns separately to examine the relationship with the meteorological fields is that existing separation methods for satellite BrO data are difficult to apply to a long-term dataset in both hemispheres. They also have large uncertainties in connection with low pressure systems and large tropopause height changes which affect both stratospheric and tropospheric columns. This study aims to investigate how the meteorological system generally affects the total BrO column, rather than separate the effects on the enhancement of BrO in each atmospheric layer."

- In section 4.1 (P8 L237):

In this study, a reference grid with 200×200 km resolution was used. The enhanced BrO occurrence frequency $f_{EBrO}$ was calculated by dividing the number of pixels classified as enhanced BrO column in a given reference grid cell by the total number of satellite pixels within the reference grid cell. "One thing to note is that cloud filtering was not applied in this study. Clouds can affect the BrO column retrieval generally in three ways: (1) the albedo effect related to the increase of the reflectivity and sensitivity for cloudy scenes, (2) the enhancement of optical light path due to multiple scattering inside clouds, and (3) the shielding effect that hides trace gases below clouds. The first two effects could increase the absorption of trace gases and lead to values greater than the actual total BrO column, while the third effect leads to an underestimation of the total BrO column (Antón and Loyola, 2011). Therefore, as clouds affect the BrO retrieval, it is necessary to consider the presence and characteristics of clouds for accurate BrO analysis. However, obtaining long-term reliable cloud products such as cloud fraction and cloud top height over the polar sea ice regions is difficult since detecting clouds and retrieving their properties over a bright snow/sea ice surface from satellite measurements are difficult (Heidinger and Stephens, 2000). Inaccurate cloud data may cause errors in statistical analysis using long-term data. Also, the difference between cloud free and cloudy conditions in polar sea ice regions is relatively small due to the bright surface (Figure 1 in Blechschmidt et al., 2016).

Based on these considerations, this study did not attempt to correct the effects of clouds on the enhancement of BrO columns."

**Technical corrections:**

P5 L139: Use proper multiple signs in here and thereafter, not letter "x".

This has been corrected in the revised manuscript.

P5 L160: Define DSCD.

As suggested, we have defined the sentence as follows (blue text):

When using a Pacific background spectrum, the retrieved differential slant columns (DSCD) need to be corrected by adding the BrO slant column over that region.

P28. Fig 1: Use consistent y limits for all four panels (e.g., 1e5). The current selections for each panel are a bit arbitrary.

As suggested, changes have been applied to Fig.1.

[Figure]

Figs. 2 and 4: The 0-degree Longitude sign and the 70-degree Latitude sign are jammed.

As suggested, changes have been applied to maps (Fig. 2, 4, 5, 7, 8, 10-16, 18-20) in the revised manuscript. Here, we have attached Fig.2 and 4.

[Figure]

Figure 2. Monthly spatial distribution of the occurrence frequency of enhanced total BrO columns over the Arctic (top left: March, top right: April) and Antarctic sea ice region (bottom left: September, bottom right: October) during the study period of 2008-2018.

[Figure]

Figure 4. Monthly sea level pressure for the mean field (left), the enhanced BrO case (middle), and sea level pressure anomalies (difference of sea level pressure between the enhanced BrO case and the mean field) (right) over the Arctic in March (upper panel) and April (lower panel).

P13 L421: Please provide the definition of DU (Dobson unit).

As suggested, we have revised the sentence as follows:

Salawitch et al. (2010) found that enhanced total BrO columns over the Hudson Bay observed by OMI are coincident with a low tropopause of ~5 km and high total $O_3$ column of ~450 DU (Dobson unit).

P20 L627: Capitalize each word; change "Geophysical research letters" to "Geophysical Research Letters."

This has been corrected in the revised manuscript.

P23 L736: Remove "n/a-n/a".

This has been corrected in the revised manuscript.

---

## Author Response (AR2)

P2 L48: Remove "Yang et al. 2008" just before "Pratt et al. 2013". This paper was cited in a very wrong position as it is 100 % against the mechanism of snowpack photochemistry.

This has been corrected in the revised manuscript.

P17 L555: change "which is affected by" to "which is a complex function of".

This has been corrected in the revised manuscript.

P18 L582-583: This sentence starting with "Also, surface wind speed shows positive ..." is too vague and I could not understand what you meant. What I can see from figure 21 is that, similar to the derived conclusion regarding the negative time-lag correlation between BrO VCDs and sea level air pressure, the total BrO VCDs in the Arctic has a slightly larger coefficient with the surface wind speed in the previous 1-2 days, though this relation in the Antarctic data (figure 22) is not as clear as in the Arctic dataset. These negative time-lags correlations between total BrO VCDs and air pressure as well as surface wind speed have a strong indication that the enhanced BrO plumes in polar regions are likely transported and are clearly synoptic system related, which should be mentioned in the summary and conclusions.

As suggested, we have revised the text in section 4.3 (P18 L582-583) and added sentences in section 5 as follows:

In section 4.3 (P18 L582-583):

Also, surface wind speed shows positive correlations with total BrO VCD, and in particular, total BrO VCD in the Arctic has a slightly larger correlation coefficient with the surface wind speed in the previous 1-2 days, which indicates the influences of transported BrO plumes.

In section 5 (P19 L623):

Regarding the time-lagging effects of meteorological factors on total BrO VCDs, the strongest correlation for the tropopause height and surface air temperature is found without the time-lag, whereas sea level pressure and surface wind speed have larger correlations with negative time-lags. These negative time-lags correlations between total BrO VCD and sea level pressure as well as surface wind speed indicate that the enhanced BrO plumes in polar regions are likely transported and are clearly synoptic system related.

P19 L611: remove the space between 50 and %.

This has been corrected in the revised manuscript.

[revised manuscript text omitted]